# FOCUS: Fairness via Agent-Awareness for Federated Learning on Heterogeneous Data

## Abstract

Federated learning (FL) provides an effective collaborative training paradigm, allowing local agents to train a global model jointly without sharing their local data to protect privacy. However, due to the heterogeneous nature of local data, it is challenging to optimize or even define *fairness* of the trained global model for the agents. For instance, existing work usually considers accuracy equity as fairness for different agents in FL, which is limited, especially under the heterogeneous setting, since it is intuitively "unfair" to enforce agents with high-quality data (e.g., hospitals with high-resolution data and fine-grained labels) to achieve similar accuracy to those who contribute low-quality data (e.g., hospitals with low-resolution data and noisy labels), which may discourage the agents from participating in FL. In this work, we aim to address such limitations and propose a formal fairness definition in FL, *fairness via agent-awareness* (FAA), which takes different contributions of heterogeneous agents into account. Under FAA, the performance of agents with high-quality data will not be sacrificed just due to the existence of large amounts of agents with low-quality data. In addition, we propose a fair FL training algorithm based on agent clustering (FOCUS) to achieve fairness in FL measured by FAA. Theoretically, we prove the convergence and optimality of FOCUS under mild conditions for linear and general convex loss functions with bounded smoothness. We also prove that FOCUS always achieves higher fairness in terms of FAA compared with standard FedAvg under both linear and general convex loss functions. Empirically, we evaluate FOCUS on four datasets, including synthetic data, images, and texts under different settings, and we show that FOCUS achieves significantly higher fairness in terms of FAA while maintaining similar or even higher prediction accuracy compared with FedAvg and other existing fair FL algorithms.

## 1 Introduction

Federated learning (FL) is emerging as a promising approach to enable scalable intelligence over distributed settings such as mobile networks (Lim et al., 2020; Hard et al., 2018). Given the wide adoption of FL, including medical analysis (Sheller et al., 2020; Adnan et al., 2022), recommendation systems (Minto et al., 2021; Anelli1 et al., 2021), and personal Internet of Things (IoT) devices (Alawadi et al., 2021), how to ensure the fairness of the trained global model in FL is of great importance before its large-scale deployment, especially when the data quality/contributions of different agents are different in the heterogeneous setting.

In general, fairness is defined as the protection of a specific attribute, and fair FL is usually in the form of equity, which means that each individual that joins collaborative learning would not suffer from bad performance due to their identity. Several studies have explored fairness in FL, which mainly focus on the fairness of the final trained model regarding the protected attributes without considering different contributions of agents (Chu et al., 2021; Hu et al., 2022) or the accuracy parity across agents (Li et al., 2020b; Donahue & Kleinberg, 2022a; Mohri et al., 2019). Some works have considered the properties of local agents, such as the local data properties (Zhang et al., 2020; Kang et al., 2019) and data size (Donahue & Kleinberg, 2022b). However, the fairness analysis in FL under heterogeneous data distributions is still lacking. Thus, in this paper, we aim to ask: *What is the fairness of FL that is able to take different contributions of heterogeneous local agents into account? Can we enhance the fairness of FL by providing advanced training algorithms?*

To better understand the fairness of FL under heterogeneous data, in this work, we aim to define and enhance fairness by explicitly considering different contributions of heterogeneous agents. In particular, for FL trained with standard FedAvg protocol (McMahan et al., 2017), if we denote the data of agent $e$ as $D_e$ with size $n_e$ and the total number of data as $n$, the final trained global model aims to minimize the loss with respect to the global distribution $\mathcal{P} = \sum_{e=1}^{E} \frac{n_e}{n} D_e$, where $E$ is the total number of agents. In practice, some local agents may have low-quality data (e.g., free riders), so intuitively it is "unfair" to train the final model regarding such global distribution over all agents, which will sacrifice the performance of agents with high-quality data. For example, considering the FL applications for medical analysis, some hospitals have high-resolution medical data and fine-grained labels, which cost a large amount of money to collect the data from advanced equipment and to crowdsource data labeling. In contrast, some hospitals may have low-resolution medical data and noisy labels. In such a setting, high-quality agents may not be willing to participate in collaborative learning with low-quality agents because they could have achieved higher accuracy by standalone local training. Therefore, a proper fairness notion is important to encourage agents to participate in FL and ensure fairness. In this paper, we define **fairness via agent-awareness in FL (FAA)** as $\mathcal{FAA}(\{\theta_e\}_{e \in [E]}) = \max_{e_1, e_2 \in E} \left| \mathcal{E}_{e_1}(\theta_{e_1}) - \mathcal{E}_{e_2}(\theta_{e_2}) \right|$, measured by the maximal *excess risk* difference between any pair of agents $e_1, e_2 \in E$. The excess risk of each agent is calculated as $\mathcal{E}_e(\theta_e) = \mathcal{L}_e(\theta_e) - \min_{\theta^*} \mathcal{L}_e(\theta^*)$, which stands for the loss of user $e$ evaluated on the FL model $\theta_e$ subtracted by the Bayes optimal error of the local data distribution (Opper & Haussler, 1991). For each agent, a lower excess risk $\mathcal{E}_e(\theta_e)$ indicates more gain from the FL model $\theta_e$ w.r.t the local distribution because its loss $\mathcal{L}_e(\theta_e)$ is more closed to its Bayes optimal error. Notably, reducing FAA enforces the *equity of excess risks* among agents, following the philosophy that each agent should *"gain the same"* from participating in FL. Therefore, lower FAA indicates stronger fairness for FL.

Based on our fairness definition FAA, we then propose a *fair FL algorithm based on agent clustering* (FOCUS) to improve the fairness of FL. Specifically, we first cluster the local agents based on their data distributions and then train a model for each cluster. During inference time, the final prediction will be the weighted aggregation over the prediction result of each model trained with the corresponding clustered local data. Theoretically, we prove that the final converged stationary point of FOCUS is exponentially close to the optimal cluster assignment under mild conditions. In addition, we prove that the fairness of FOCUS in terms of FAA is strictly higher than that of the standard FedAvg under both linear models and general convex losses. Empirically, we evaluate FOCUS on four datasets, including synthetic data, images, and texts, and we show that FOCUS achieves higher fairness measured by FAA than FedAvg and SOTA fair FL algorithms while maintaining similar or even higher prediction accuracy.

**Technical contributions**. In this work, we define and improve FL fairness in heterogeneous settings by considering different contributions of heterogeneous local agents. We make contributions on theoretical and empirical fronts.

- We formally define *fairness via agent-awareness (FAA)* in FL based on agent-level excess risks to measure fairness in FL, and explicitly take the heterogeneity nature of local agents into account.
- We propose a fair FL algorithm via agent clustering (FOCUS) to improve fairness measured by FAA, especially in the heterogeneous setting. We prove the convergence rate and optimality of FOCUS under linear models and general convex losses.
- We prove that FOCUS achieves stronger fairness measured by FAA compared with FedAvg for both linear models and general convex losses.
- Empirically, we compare FOCUS with FedAvg and SOTA fair FL algorithms on four datasets, including synthetic data, images, and texts under heterogeneous settings. We show that FOCUS indeed achieves stronger fairness measured by FAA while maintaining similar or even higher prediction accuracy on all datasets.

## 2 RELATED WORK

**Fair Federated Learning**    There have been several studies exploring fairness in FL. Li et al. (2020b) first define agent-level fairness by considering *accuracy equity* across agents and achieve fairness by assigning the agents with worse performance with higher aggregation weight during training. However, such a definition of fairness fails to capture the heterogeneous nature of local agents. Mohri et al. (2019) pursue accuracy parity by improving the performance of the worst-performing agent. Wang et al. (2021) propose to mitigate conflict gradients from local agents to enhance fairness. Instead of pursuing fairness with one single global model, Li et al. (2021) propose to train a personalized model for each agent to achieve accuracy equity for the personalized models.

Zhang et al. (2020) predefine the agent contribution levels based on an oracle assumption (e.g., data volume, data collection cost, etc.) for fairness optimization, which lacks quantitative measurement metrics in practice. Xu et al. (2021) approximate the Shapely Value based on gradient cosine similarity to evaluate agent contribution. However, Zhang et al. (2020) point out that Shapely Value may discourage agents with rare data, especially under heterogeneous settings. Here we provide an algorithm to quantitatively measure the contribution of local data based on each agent's excess risk, which will not be affected even if the agent is the minority.

**Clustered Federated Learning** Clustered FL algorithms are initially designed for multitasking and personalized federated learning, which assumes that agents can be naturally partitioned into clusters (Ghosh et al., 2020; Xie et al., 2021; Sattler et al., 2021; Marfoq et al., 2021). Existing clustering algorithms usually aim to assign each agent to a cluster that provides the lowest loss (Ghosh et al., 2020), optimize the clustering center to be close to the local model (Xie et al., 2021), or cluster agents with similar gradient updates (with respect to, e.g., cosine similarity (Sattler et al., 2021)) to the same cluster. In addition to these hard clustering approaches (i.e., each agent only belongs to one cluster), soft clustering has also been studied (Marfoq et al., 2021; Li et al., 2022; Ruan & Joe-Wong, 2022; Stallmann & Wilbik, 2022), which enables the agents to benefit from multiple clusters. However, none of these works considers the fairness of clustered FL and the potential implications, and our work makes the first attempt to bridge them.

## 3 FAIR FEDERATED LEARNING ON HETEROGENEOUS DATA

In this section, we first define our fairness via agent-awareness (FAA) in FL with heterogeneous data and then introduce our fair FL based on the agent clustering (FOCUS) algorithm to achieve FAA.

### 3.1 FAIRNESS VIA AGENT-AWARENESS IN FL (FAA) WITH HETEROGENEOUS DATA

Given a set of $E$ agents participated in the FL training process, each agent $e$ only has access to its local dataset: $D_e = \{(x_e, y_e)\}_{i=1}^{n_e}$, which is sampled from a distribution $\mathcal{P}_e$. The goal of standard FedAvg training is to minimize the overall loss $\mathcal{L}_E(\theta)$ based on the local loss $\mathcal{L}_e(\theta)$ of each agent:

$$\min_\theta \mathcal{L}_E(\theta) = \sum_{e \in [E]} \frac{|D_e|}{n} \mathcal{L}_e(\theta), \qquad \mathcal{L}_e(\theta) = \mathbb{E}_{(x,y) \in \mathcal{P}_e} \ell(h_\theta(x), y). \tag{1}$$

where $\ell(\cdot, \cdot)$ is a loss function given model prediction $h_\theta(x)$ and label $y$ (e.g., cross-entropy loss), $n = \sum_{e \in [E]} |D_e|$ represents the total number of training samples, and $\theta$ represents the parameter of trained global model.

Intuitively, the performance of agents with high-quality data (e.g., clean or better generality) could be severely compromised by the existence of large amounts of agents with low-quality data (e.g., noisy or lower generality) under FedAvg. To solve such a problem and characterize the distinctions of local data distributions (contributions) among agents to ensure fairness, we propose fairness via agent-awareness in FL (FAA) as below.

**Definition 1 (Fairness via agent-awareness for FL (FAA)).** *Given a set of agents $E$ in FL, the overall fairness score among all agents is defined as the maximal difference of excess risks for any pair of agents:*

$$\mathcal{FAA}(\{\theta_e\}_{e \in [E]}) = \max_{e_1, e_2 \in [E]} \left| \mathcal{E}_{e_1}(\theta_{e_1}) - \mathcal{E}_{e_2}(\theta_{e_2}) \right|. \tag{2}$$

*where $\theta_e$ is the local model for agent $e \in [E]$. The excess risk $\mathcal{E}_e(\theta_e)$ for agent $e$ given model $\theta_e$ is defined as the difference between the population loss $\mathcal{L}_e(\theta_e)$ and the Bayes optimal error of the corresponding data distribution, i.e.,*

$$\mathcal{E}_e(\theta_e) = \mathcal{L}_e(\theta_e) - \min_{\theta^*} \mathcal{L}_e(\theta^*), \tag{3}$$

*where $\theta^*$ denotes any possible models.*

Note that in FedAvg, each client uses the global model $\theta$ as its local model $\theta_e$. Definition 1 represents a quantitative data-dependent measurement of agent-level fairness. Instead of forcing accuracy equity among all agents regardless of their data distributions, we define agent-level fairness as the equity of *excess risks* among agents, which takes the contributions of local data into account by measuring their Bayes errors. For instance, when a local agent has low-quality data, although the corresponding utility loss would be high, the Bayes error of such low-quality data is also high, and thus the excess risk of the user is still low, enabling the agents with high-quality data to achieve low utility loss for fairness. According to the definition, we note that **lower FAA indicates stronger fairness among agents**.

## 3.2 FAIR FEDERATED LEARNING ON HETEROGENEOUS DATA VIA CLUSTERING (FOCUS)

**Method Overview.** To enhance the fairness of FL in terms of FAA, we provide an agent clustering-based FL algorithm (FOCUS) by partitioning agents conditioned on their data distributions. Intuitively, grouping agents with similar data distributions together helps to improve fairness, since it reduces the intra-cluster data heterogeneity. We will analyze the fairness achieved by FOCUS and compare it with standard FedAvg both theoretically (Section 4.2) and empirically (Section 5).

Our FOCUS algorithm (Algorithm 1) leverages the Expectation-Maximization algorithm to perform agent clustering. Define $M$ as the number of clusters and $E$ as the number of agents. The goal of FOCUS is to simultaneously optimize the soft clustering labels $\Pi$ and model weights $W$. Specifically, $\Pi = \{\pi_{em}\}_{e \in [E], m \in [M]}$ are the dynamic soft clustering labels, representing the estimated probability that agent $e$ belongs to cluster $m$; $W = \{w_m\}_{m \in [M]}$ represent the model weights for $M$ data clusters. Given $E$ agents with datasets $D_1, \dots, D_E$, our FOCUS algorithm follows a two-step scheme that alternately optimizes $\Pi$ and $W$.

**E step.** Expectation steps update the cluster labels $\Pi$ given the current estimation of $(\Pi, W)$. At $k$-th communication round, the server broadcasts the $M$ cluster models to all agents. The agents calculate the expected training loss $\mathbb{E}_{(x,y) \in D_e} \ell(x, y; w_m^{(t)})$ for each cluster model $w_m^{(t)}$, $m \in [M]$, and then update the soft clustering labels $\Pi$ according to Eq. (8).

**M step.** The goal of M steps in Eq. (9) is to minimize a weighted sum of empirical losses for all local agents. However, given distributed data, it is impossible to find its exact optimal solution in practice. Thus, we specify a concrete protocol in Eq. (4) $\sim$ Eq. (6) to estimate the objective in Eq. (9). At $t$-th communication round, for each cluster model $w_m^{(t)}$ received from server, each agent $e$ first initializes its local model $\theta_{em(0)}^{(t)}$ as $w_m^{(t)}$, and then updates the model using its own dataset. To reduce communication costs, each agent is allowed to run SGD locally for $K$ local steps as shown in Eq. (5). After $K$ local steps, each agent sends the updated models $\theta_{em(K)}^{(t)}$ back to the central server, and the server aggregates the models of all agents by a weighted average based on the soft clustering labels $\{\pi_{em}\}$. We provide theoretical analysis for the convergence and optimality of FOCUS considering these multiple local updates in Section 4.

$$\text{Clients:} \quad \theta_{em(0)}^{(t)} = w_m^{(t)}. \tag{4}$$

$$\theta_{em(k+1)}^{(t)} = \theta_{em(k)}^{(t)} - \eta_k \nabla \sum_{i=1}^{n_e} \ell\left(h_{\theta_{em(k)}^{(t)}}(x_e^{(i)}), y_e^{(i)}\right), \forall k = 1, \dots, K-1. \tag{5}$$

$$\text{Server:} \quad w_m^{(t+1)} = \sum_{e=1}^{E} \frac{\pi_{em}^{(t+1)} \theta_{em(K)}^{(t)}}{\sum_{e'=1}^{E} \pi_{e'm}^{(t+1)}}. \tag{6}$$

**Inference.** At inference time, each agent ensembles the $M$ models by a weighted average on their prediction probabilities, i.e., a agent $e$ predicts $\sum_{m=1}^{M} \pi_{em} h_{w_m}(x)$ for input $x$. Suppose a test dataset $D_e^{test}$ is sampled from distribution $\mathcal{P}_e$. The test loss can be calculated by

$$\mathcal{L}_{test}(W, \Pi) = \frac{1}{|D_e^{test}|} \sum_{(x,y) \in D_e^{test}} \ell\left(\sum_{m=1}^{M} \pi_{em} h_w(x), y\right) \tag{7}$$

For unseen agents that do not participate in the training process, their clustering labels $\Pi$ are unknown. Therefore, an unseen agent $e$ computes its one-shot clustering label $\pi_{em}^{(1)}, m \in [M]$ according to Eq. (8), and outputs predictions $\sum_{m=1}^{M} \pi_{em}^{(1)} h_{w_m}(x)$ for the test sample $x$.

## 4 THEORETICAL ANALYSIS OF FOCUS

In this section, we first present the convergence and optimality guarantees of our FOCUS algorithm; and then prove that it improves the fairness of FL regarding FAA. Our analysis considers linear models and then extends to nonlinear models with smooth and strongly convex loss functions.

### 4.1 CONVERGENCE ANALYSIS

**Linear models.** We first start with linear models for analysis simplicity. Suppose there are $E$ agents, each with a local dataset $D_e = \{(x_e^{(i)}, y_e^{(i)})\}_{i=1}^{n_e}, (e \in [E])$ generated from a Gaussian distribution. Specifically, we assume each dataset $D_e$ has a mean vector $\mu_e \in \mathbb{R}^d$, and $(x_e^{(i)}, y_e^{(i)})$ is

---

**Algorithm 1** EM clustered federated learning algorithm

---

**Input:** Data $D_1, \ldots, D_E$; $E$ remote agents and $M$ learning models.

Initialize weights $w_m^{(0)}$ and $\pi_{em}^{(0)} = \frac{1}{M}$ for $m \in [M]$ and $e \in [E]$.

**for** $t = 0$ to $T - 1$ **do**

    **for** agent $e \in [E]$ **do**

        **for** model $m \in [M]$ **do**

            E step:
$$\pi_{em}^{(t+1)} \leftarrow \frac{\pi_{em}^{(t)} \exp\left(-\mathbb{E}_{(x,y) \in D_e} \ell(x, y; w_m^{(t)})\right)}{\sum_{m=1}^{M} \pi_{em}^{(t)} \exp\left(-\mathbb{E}_{(x,y) \in D_e} \ell(x, y; w_m^{(t)})\right)} \tag{8}$$

        **end for**

    **end for**

    **for** model $m \in [M]$ **do**

        M step:
$$w_m^{(t+1)} \leftarrow \arg\min_w \sum_{e=1}^{E} \pi_{em}^{(t+1)} \sum_{i=1}^{n_e} \ell\left(h_w(x_e^{(i)}), y_e^{(i)}\right) \tag{9}$$

    **end for**

**end for**

**return** model weights $w_m^{(T)}$

---

generated by $y_e^{(i)} = \mu_e^T x_e^{(i)} + \epsilon_e^{(i)}$, where $x_e^{(i)}$ is a random vector $x_e^{(i)} \sim \mathcal{N}(0, \delta^2 I_d)$ and the label $y_e^{(i)}$ is blurred by some random noise $\epsilon_e^{(i)} \sim \mathcal{N}(0, \sigma^2)$. Each agent is asked to minimize the mean squared error to estimate $\mu_e$, so the empirical loss function for a local agent given dataset $D_e$ is

$$\mathcal{L}_{emp}(D_e; w) = \frac{1}{n_e} \sum_{i=1}^{n_e} (w^T x_e^{(i)} - y_e^{(i)})^2. \tag{10}$$

We further make the following assumption about the heterogeneous agents.

**Assumption 1** (Separable distributions). *Suppose there are $M$ predefined vectors $\{w_i^*\}_{i=1}^M$, where for any $m_1, m_2 \in [M]$, $\|w_{m_1}^* - w_{m_2}^*\|_2 \geq R$. A set of agents $E$ satisfy separable distributions if they can be divided into $M$ subsets $S_1, \ldots, S_M$ such that, for any agent $e \in S_m$, $\|\mu_e - w_m^*\|_2 \leq r < \frac{R}{2}$.* Assumption 1 guarantees that the heterogeneous local data distributions are separable so that an optimal clustering solution exists, in which $\{w_1^*, \ldots, w_M^*\}$ are the centers of clusters.

We next present Theorem 1 to demonstrate the linear convergence rate to the optimal cluster centers for FOCUS. Detailed proofs can be found in Appendix B.1.

**Theorem 1.** *Consider the agent set $E$ satisfying separable distributions as assumption 1. Given trained $M$ models and $\forall e, m$, $\pi_{em}^{(0)} = \frac{1}{M}$. Under the natural initialization $w_m$ for each model $m \in [M]$, which satisfies $\exists \Delta_0 > 0, \|w_m^{(0)} - w_m^*\|_2 \leq \min_{m' \neq m} \|w_m^{(0)} - w_{m'}^*\|_2 - 2(r + \Delta_0)$ and $|D_e| = O(d)$. If learning rate $\eta \leq \min(\frac{1}{4\delta^2}, \frac{\beta}{\sqrt{T}})$, FOCUS converges by*

$$\pi_{em}^{(T)} \geq \frac{1}{1 + (M-1) \cdot \exp(-2R\delta^2 \Delta_0 T)}, \forall e \in S_m \tag{11}$$

$$\mathbb{E}\|w_m^{(T)} - w_m^*\|_2^2 \leq (1 - \frac{2\eta \gamma_m \delta^2}{M})^{KT} (\|w_m^{(0)} - w_m^*\|_2^2 + A) + 2MKr + \frac{M\delta^2 E\beta}{2\sqrt{T}} O(K^3, \sigma^2). \tag{12}$$

*where $T$ is the total number of communication rounds; $K$ is the number of local updates in each communication round; $\gamma_m = |S_m|$ is the number of agents in the $m$-th cluster, and*

$$A = \frac{2EK(M-1)\delta^2}{(1 - \frac{2\eta \delta^2 \gamma_m}{M})^K - \exp(-2R\delta^2 \Delta_0)} \quad \text{(caused by initial inaccurate clustering).} \tag{13}$$

*Proof sketch.* To prove this theorem, we first consider E steps and M steps separately to derive corresponding convergence lemmas (Lemmas 1 and 2). In E steps, the soft cluster labels $\pi_{em}$ increase for all $e \in S_m$, as long as $\|w_m^{(t)} - w_m^*\|_2 < \|w_{m'}^{(t)} - w_m^*\|_2, \forall m' \neq m$. On the other hand, $\|w_m^{(t)} - w_m^*\|$ is guaranteed to shrink linearly as long as $\pi_{em}$ is large enough for any $e \in S_m$. We then integrate Lemmas 1 and 2 and prove Theorem 1 using an induction statement. □

**Remarks.** Theorem 1 shows the convergence of parameters $(\Pi, W)$ to a near-optimal solution. Eq. (11) implies that the agents will be *correctly clustered* since $\pi_{em}$ will converge to 1 as the number of communication rounds $K$ increases. In Eq. (12), the first term diminishes exponentially, while the second term $2MKr$ reflects the intra-cluster distribution divergence $r$. The last term originates

from the data heterogeneity among clients across different clusters. Its influence is amplified by the number of local updates ($O(K^3)$) and will also diminish to zero as the number of communication rounds $T$ goes to infinity. Our convergence analysis is conditioned on the natural clustering initialization for model weights $w_m^{(0)}$ towards a corresponding cluster center $w_m^*$, which is standard in convergence analysis for a mixture of models (Yan et al., 2017; Balakrishnan et al., 2017).

**Smooth and strongly convex loss functions.** Next, we extend our analysis to a more general case of non-linear models with $L$-smooth and $\mu$-strongly convex loss function.

**Assumption 2** (Smooth and strongly convex loss functions). *The population loss functions $\mathcal{L}_e(\theta)$ for each agent $e$ is $L$-smooth, i.e., $\|\nabla^2 \mathcal{L}_e(\theta)\|_2 \leq L$. The loss functions are $\mu$-strongly convex, if the eigenvalues $\lambda$ of the Hessian matrix $\nabla^2 \mathcal{L}_e(\theta)$ satisfy $\lambda_{\min}(\nabla^2 \mathcal{L}_e(\theta)) \geq \mu$.*

We further make an assumption similar to Assumption 1 following the similar philosophy.

**Assumption 3** (Separable distributions). *A set of agents $E$ satisfy separable distributions if they can be partitioned into $M$ subsets $S_1, \ldots, S_M$ with $w_1^*, ..., w_M^*$ representing the center of each set respectively, and the optimal parameter $\theta^*$ of each local loss $\mathcal{L}_e$ (i.e., $\theta_e^* = \arg\min_\theta \mathcal{L}_e(\theta)$) satisfy*

$$\|\theta_e^* - w_m^*\|_2 \leq r \tag{14}$$

*In the meantime, agents from different subsets have different data distributions, such that*

$$\|w_{m_1}^* - w_{m_2}^*\|_2 \geq R, \forall m_1, m_2 \in [M], m_1 \neq m_2. \tag{15}$$

**Theorem 2.** *Consider the agent set $E$ satisfying separable distributions as assumption 3. Suppose loss functions have bounded variance for gradients on local datasets, i.e., $\mathbb{E}_{(x,y)\sim\mathcal{D}_e}[\|\nabla\ell(x,y;\theta) - \nabla\mathcal{L}_e(\theta)\|_2^2] \leq \sigma^2$, and the population losses are bounded, i.e., $\mathcal{L}_e \leq G, \forall e \in [E]$. If let $\pi_{em}^{(0)} = \frac{1}{M}, \exists \Delta_0 > 0, \|w_m^{(0)} - w_m^*\|_2 \leq \frac{\sqrt{\mu}R}{\sqrt{\mu}+\sqrt{L}} - r - \Delta_0$, and the learning rate of each agent $\eta \leq \min(\frac{1}{2(\mu+L)}, \frac{\beta}{\sqrt{T}})$, FOCUS converges by*

$$\pi_{em}^{(T)} \geq \frac{1}{1 + (M-1)\exp(-\mu R\Delta_0 T)}, \ \forall e \in S_m \tag{16}$$

$$\mathbb{E}\|w_m^{(T)} - w_m^*\|_2^2 \leq (1 - \eta A)^{KT}(\|w_m^{(0)} - w_m^*\|_2^2 + B) + O(Kr) + \frac{ME\beta O(K^3, \frac{\sigma^2}{n_e})}{\sqrt{T}} \tag{17}$$

*where $T$ is the total number of communication rounds; $K$ is the number of local updates in each communication round; $\gamma_m = |S_m|$ is the number of agents in the $m$-th cluster, and*

$$\underbrace{A = \frac{2\gamma_m}{M}\frac{\mu L}{\mu + L}}_{\text{related to convergence rate}}, \underbrace{B = \frac{GMTE(\frac{4L}{\mu} + \frac{6}{\mu(\mu+L)})}{(1 - \eta A)^K - \exp(-\mu R\Delta_0)}}_{\text{caused by the offset of initial clustering}}. \tag{18}$$

*Proof sketch.* We analyze the evolution of parameters $(\Pi, W)$ for E steps in Lemma 3 and M steps in Lemma 4. Lemma 3 shows that the soft cluster labels $\pi_{em}$ increase for all $e \in S_m$ in E steps as long as $\|w_m - w_m^*\|_2 < \frac{\sqrt{\mu}R}{\sqrt{\mu}+\sqrt{L}} - r$; whereas Lemma 4 guarantees that the model weights $w_m$ get closer to the optimal solution $w_m^*$ in M steps. We combine Lemmas 3 and 4 by induction to prove this theorem. Detailed proofs are deferred to Appendix B.2.3. □

**Remarks.** Theorem 2 extends the convergence guarantee of $(\Pi, W)$ from linear models (Theorem 1) to general models with smooth and convex loss functions. For any agent $e$ that belongs to a cluster $m$ ($e \in S_m$), its soft cluster label $\pi_{em}$ converges to 1 based on Eq. (16), indicating the clustering optimality. Meanwhile, the model weights $W$ converge linearly to a near-optimal solution. The error term $O(Kr)$ in Eq. (17) is expected, since $r$ represents the data divergence within each cluster and $w_m^*$ denotes the center of each cluster. The last term in Eq. (17) implies a trade-off between communication cost and convergence speed. Increasing $K$ reduces communication cost by $O(\frac{1}{K})$ but at the expanse of slowing down the convergence.

## 4.2 FAIRNESS ANALYSIS

To theoretically show that FOCUS achieves stronger fairness in FL based on FAA, here we focus on a simple yet representative case where all agents share similar distributions except one outlier agent.

**Linear models.** We first concretize such a scenario for linear models. Suppose we have $E$ agents learning weights for $M$ linear models. Their local data $D_e(e \in [E])$ are generated by $y_e^{(i)} = \mu_e^T x_e^{(i)} - \epsilon_e^{(i)}$ with $x_e^{(i)} \sim \mathcal{N}(0, \delta^2 I_d)$ and $\epsilon_e^{(i)} \sim \mathcal{N}(0, \sigma_e^2)$. $E - 1$ agents learn from normal dataset with ground truth vector $\mu_1, \ldots, \mu_{E-1}$ and $\|\mu_e - \mu^*\|_2 \leq r$, while the $E$-th agent has an outlier data distribution, with its the ground truth vector $\mu_E$ far away from other agents, i.e., $\|\mu_E - \mu^*\|_2 \geq R$.

As stated in Theorem 1, the soft clustering labels and model weights $(\Pi, W)$ converge linearly to the global optimum. Therefore, we analyze the fairness of FOCUS, assuming an optimal $(\Pi, W)$ is reached. We compare the FAA achieved by FOCUS and FedAvg to underscore how our algorithm helps improve fairness for heterogeneous agents.

**Theorem 3.** *When a single agent has an outlier distribution, the fairness FAA achieved by FOCUS algorithm with two clusters $M = 2$ is*

$$\mathcal{FAA}_{focus}(W, \Pi) \leq \delta^2 r^2. \tag{19}$$

*while the fairness FAA achieved by FedAvg is*

$$\mathcal{FAA}_{avg}(W) \geq \delta^2 \Big( \frac{R^2(E-2) - 2Rr}{E} + r^2 \Big) = \Omega(\delta^2 R^2). \tag{20}$$

**Remarks.** When a single outlier exists, the fairness gap between Fedavg and FOCUS is shown by Theorem 3.

$$\mathcal{FAA}_{avg}(W) - \mathcal{FAA}_{focus}(W, \Pi) \geq \delta^2 \Big( \frac{R^2(E-2) - 2Rr}{E} \Big). \tag{21}$$

As long as $R > \frac{2r}{E-2}$, FOCUS is guaranteed to achieve stronger fairness (i.e., lower FAA) than FedAvg. Note that *the outlier assumption only makes sense when $E > 2$* since one cannot tell which agent is the outlier when $E = 2$. Also, we naturally assume $R > 2r$ so that the two underlying clusters are at least separable. Therefore, we conclude that FOCUS dominates than FedAvg in terms of FAA. Here we only discuss the scenario of a single outlier agent for clarity, but similar conclusions can be drawn for multiple underlying clusters and $M > 2$, as discussed in Appendix C.1.

**Smooth and strongly convex loss functions.** We generalize the fairness analysis to nonlinear models with smooth and convex loss functions. To illustrate the superiority of our FOCUS algorithms in terms of FAA fairness, we similarly consider training in the presence of an outlier agent. Suppose we have $E$ agents that learn weights for $M$ models. We assume their population loss functions are $L$-smooth, $\mu$-strongly convex (as in Assumption 2) and bounded, i.e., $\mathcal{L}_e(\theta) \leq G$. $E - 1$ agents learn from similar data distributions, such that the total variation distance between the distributions of any two different agents $i, j \in [E - 1]$ is no greater than $r$: $D_{TV}(\mathcal{P}_i, \mathcal{P}_j) \leq r$. On the other hand, the $E$-th agent has an outlier data distribution, such that the Bayes error $\mathcal{L}_E(\theta_i^*) - \mathcal{L}_E(\theta_E^*) \geq R$ for any $i \in [E - 1]$. We claim that this assumption can be reduced to a lower bound on H-divergence (Zhao et al., 2022) between distributions $\mathcal{P}_i$ and $\mathcal{P}_E$ that $D_H(\mathcal{P}_i, \mathcal{P}_E) \geq \frac{LR}{4\mu}$. (See proofs in Appendix C.3.)

**Theorem 4.** *The fairness FAA achieved by FOCUS with two clusters $M = 2$ is*

$$\mathcal{FAA}_{focus}(W, \Pi) \leq \frac{2Gr}{E-1} \tag{22}$$

*Let $B = \frac{2Gr}{E-1}$. The fairness achieved by FedAvg is*

$$\mathcal{FAA}_{avg}(W) \geq \Big( \frac{E-1}{E} - \frac{L}{\mu E^2} \Big) R - \Big( 1 + \frac{L(E-1)}{\mu E} - \frac{L^2}{\mu^2 E} \Big) B - \frac{2L}{\mu E} \sqrt{B(R - \frac{L}{\mu}B)} \tag{23}$$

**Remarks.** Notably, when the outlier distribution is very different from the normal distribution, such that $R \gg Gr$ (which means $B \ll R$), we simplify Eq. (23) as

$$\mathcal{FAA}_{avg}(W) \geq ( \frac{E-1}{E} - \frac{L}{\mu E^2} )R. \tag{24}$$

Note that $\mathcal{FAA}_{focus}(W, \Pi) \leq B \ll R$, so the fairness FAA achieved by FedAvg is always larger (weaker) than that of FOCUS, as long as $E \geq \sqrt{L/\mu}$, indicating the effectiveness of FOCUS.

## 5 EXPERIMENTAL EVALUATION

We conduct extensive experiments on various heterogeneous data settings to evaluate the fairness measured by FAA for FOCUS, FedAvg (McMahan et al., 2017), and two baseline fair FL algorithms (i.e., q-FFL (Li et al., 2020b) and AFL (Mohri et al., 2019)). We show that FOCUS achieves significantly higher fairness measured by FAA while maintaining similar or even higher accuracy.

## 5.1 EXPERIMENTAL SETUP

**Data and Models.** We carry out experiments on four different datasets with heterogeneous data settings, ranging from synthetic data for linear models to images (rotated MNIST (Deng, 2012) and rotated CIFAR (Krizhevsky, 2009)) to text data for sentiment classification on Yelp (Zhang et al., 2015) and IMDb (Maas et al., 2011) datasets. We train a fully connected model consisting of two linear layers with ReLU activations for MNIST, a ResNet 18 model (He et al., 2016) for CIFAR, and a pre-trained BERT-base model (Devlin et al., 2019) for the text data. We refer the readers to Appendix D for more implementation details.

**Evaluation Metrics and Implementation Details.** We consider three evaluation metrics: average test accuracy, average test loss, FAA for fairness, and the existing fairness metric "agnostic loss" introduced by Mohri et al. (2019). For FedAvg, we evaluate the trained global model on each agent's test data; for FOCUS, we train $M$ models corresponding to $M$ clusters, and use the soft clustering labels $\Pi = \{\pi_{em}\}_{e \in [E], m \in [M]}$ to make aggregated predictions on each agent's test data. We also report the performance of existing fair FL algorithms (i.e., q-FFL (Li et al., 2020b), AFL (Mohri et al., 2019), Ditto (Li et al., 2021), and CGSV Xu et al. (2021)) as well as existing state-of-the-art FL algorithms in heterogeneous data settings (i.e., FedMA (Wang et al., 2020), Bayesian nonparametric FL (Yurochkin et al., 2019) and FedProx (Li et al., 2020a) in Appendix D.2).

To evaluate FAA of different algorithms, we estimate the Bayes optimal loss $\min_w \mathcal{L}_e(w)$ for each local agent $e$. Specifically, we train a centralized model based on the subset of agents with similar data distributions (i.e., the same ground-truth cluster) and use it as a *surrogate* to approximate the Bayes optimum. We select the agent pair with the maximal difference of excess risks to measure fairness in terms of FAA calculated following Definition 1.

## 5.2 EVALUATION RESULTS

**Synthetic data for linear models.** We first evaluate FOCUS on linear regression models with synthetic datasets. We set up $E = 10$ agents with data sampled from Gaussian distributions. Each agent $e$ is assigned with a local dataset of $D_e = \{(x_e^{(i)}, y_e^{(i)})\}_{i=1}^{n_e}$ generated by $y_e^{(i)} = \mu_e^T x_e^{(i)} + \epsilon_e^{(i)}$ with $x_e^{(i)} \sim \mathcal{N}(0, I_d)$ and $\epsilon_e^{(i)} \sim \mathcal{N}(0, \sigma^2)$. We study the case considered in Section 4.2 where a single agent has an outlier data distribution. We set the intra-cluster distance $r = 0.01$ and the inter-cluster distance $R = 1$ in our experiment. Note that it is a regression task, so we mainly report the average test loss instead of accuracy here. Table 1 shows that FOCUS achieves FAA of 0.001, which is much lower than the FAA 0.958 achieved by FedAvg, 0.699 by q-FFL, and 0.780 by AFL.

Table 1: Comparison of FOCUS, FedAvg, and fair FL algorithms q-FFL, AFL, Ditto and CGSV, in terms of average test accuracy (Avg Acc), average test loss (Avg Loss), fairness FAA and existing fairness metric Agnostic loss. FOCUS achieves the best fairness measured by FAA compared with all baselines.

| | | FOCUS | FedAvg | q-FFL $q=0.1$ | q-FFL $q=1$ | q-FFL $q=10$ | AFL | Ditto | CGSV |
|---|---|---|---|---|---|---|---|---|---|
| Synthetic | Avg Loss | **0.010** | 0.108 | 0.106 | 0.102 | 0.110 | 0.104 | 0.023 | 0.260 |
| | FAA | **0.001** | 0.958 | 0.769 | 0.717 | 0.699 | 0.780 | 0.012 | 0.010 |
| Rotated MNIST | Avg Acc | **0.953** | 0.929 | 0.922 | 0.861 | 0.685 | 0.885 | 0.940 | 0.938 |
| | Avg Loss | **0.152** | 0.246 | 0.269 | 0.489 | 1.084 | 0.429 | 0.210 | 0.222 |
| | FAA | **0.094** | 0.363 | 0.388 | 0.612 | 0.253 | 0.220 | 0.104 | 0.210 |
| | Agnostic Loss | **0.224** | 0.616 | 0.656 | 1.018 | 1.271 | 0.548 | 0.354 | 0.331 |
| Rotated CIFAR | Avg Acc | 0.929 | 0.908 | 0.897 | 0.833 | 0.565 | 0.866 | **0.932** | 0.861 |
| | Avg Loss | 0.217 | 0.262 | 0.306 | 0.704 | 1.263 | 0.324 | **0.202** | 0.511 |
| | FAA | **0.365** | 0.537 | 0.661 | 0.542 | 0.421 | 0.619 | 0.424 | 1.775 |
| | Agnostic Loss | **0.503** | 1.238 | 0.932 | 0.905 | 1.413 | 0.667 | 0.598 | 0.904 |
| Yelp/IMDb | Avg Acc | **0.940** | **0.940** | 0.938 | 0.938 | 0.909 | 0.934 | 0.933 | 0.701 |
| | Avg Loss | **0.174** | 0.236 | 0.188 | 0.179 | 0.264 | 0.187 | 0.191 | 0.547 |
| | FAA | **0.047** | 0.098 | 0.052 | 0.051 | 0.070 | 0.049 | 0.049 | 0.462 |
| | Agnostic Loss | 0.257 | 0.349 | 0.266 | 0.253 | **0.242** | 0.253 | 0.263 | 0.700 |

**Rotated MNIST and CIFAR.** Following (Ghosh et al., 2020), we rotate the images MNIST and CIFAR datasets with different degrees to create data heterogeneity among agents. Both datasets are evenly split into 10 subsets for 10 agents. For MNIST, two subsets are rotated for 90 degrees, one subset is rotated for 180 degrees, and the rest seven subsets are unchanged, yielding an FL setup with three ground-truth clusters. Similarly, for CIFAR, we rotate the images of 3 subsets for 180 degrees, thus creating two ground-truth clusters. From Table 1, we observe that FOCUS consistently achieves higher average test accuracy, lower average test loss, and lower FAA than other methods on

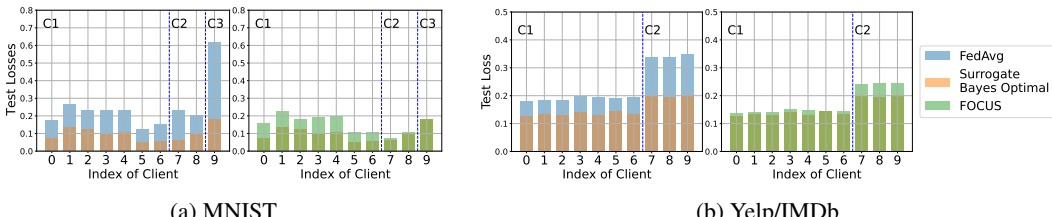

(a) MNIST            (b) Yelp/IMDb

Figure 1: The excess risk of different agents trained with FedAvg and FOCUS on MNIST (a) and Yelp/IMDb text data (b). $C_i$ denotes $i$th cluster.

both datasets. In addition, although existing fair FL algorithms q-FFL and AFL achieve lower FAA scores than FedAvg, their average test accuracy drops significantly. This is mainly because these fair algorithms are designed for performance parity via improving low-quality agents (i.e., agents with high training loss), thus sacrificing the accuracy of high-quality agents. Notably, FOCUS both improves the FAA fairness and preserves high test accuracy.

Next, we analyze the surrogate excess risk of every agent on MNIST in Fig. 1 (a). We observe that the global model trained by FedAvg obtains the highest test loss as 0.61 on the outlier cluster, which rotates 180 degrees (i.e., cluster C3), resulting in high excess risk for the 9th agent. Moreover, the low-quality data of the outlier cluster affect the agents in the 1st cluster via FedAvg training, which leads to a much higher excess risk than that of FOCUS. On the other hand, FOCUS successfully identifies clusters of the outlier distributions, i.e., clusters 2 and 3, rendering models trained from the outlier clusters independent from the normal cluster 1. As shown in Fig. 1, our FOCUS reduces the excess risks of all agents, especially for the outliers, on different datasets. This leads to strong fairness among agents in terms of FAA. Similar trends are also observed in CIFAR, in which our FOCUS reduces the surrogate excess risk for the 9th agent from 2.74 to 0.44. We omit the loss histogram of CIFAR to Appendix D.

In addition, we evaluate different numbers of outliers in Table 2. In the presence of 1, 3, and 5 outlier agents, forming 2, 3, or 4 underlying true clusters, FOCUS consistently achieves a lower FAA score and higher accuracy than baseline FedAvg.

Table 2: Comparison of FOCUS and FedAvg with different numbers of outlier agents ($k$) in terms of average test accuracy (Avg Acc) and fairness FAA.

|  |  | Rotated MNIST | | | Rotated CIFAR | | |
|---|---|---|---|---|---|---|---|
|  |  | $k=1$ | $k=3$ | $k=5$ | $k=1$ | $k=3$ | $k=5$ |
| Avg Acc | FOCUS | **0.957** | **0.953** | **0.948** | **0.939** | **0.929** | **0.872** |
|  | FedAvg | 0.945 | 0.929 | 0.910 | 0.803 | 0.908 | 0.857 |
| FAA | FOCUS | **0.159** | **0.094** | **0.153** | **1.739** | **0.365** | **1.403** |
|  | FedAvg | 0.515 | 0.363 | 0.476 | 3.456 | 0.537 | 1.848 |

In practice, we do not know the number of underlying clusters, so we set $M = 2, 3, 4$ while we have 3 true underlying clusters in Table 9 on MNIST. It shows that under different $M$, FOCUS achieves similar accuracy and fairness (see more discussion in Appendix A).

**Sentiment classification.** We evaluate FOCUS on the sentiment classification task with text data, Yelp (restaurant reviews), and IMDb (movie reviews), which naturally form data heterogeneity among 10 agents and thus create 2 clusters. Specifically, we sample 56k reviews from Yelp datasets distributed among seven agents and use the whole 25k IMDB datasets distributed among three agents to simulate the heterogeneous setting. From Table 1, we can see that while the average test accuracy of FOCUS, FedAvg, and other fair FL algorithms are similar, FOCUS achieves a lower average test loss. In addition, the FAA of FOCUS is significantly lower than other baselines, indicating stronger fairness. We also observe from Fig. 1 (b) that the excess risk of FOCUS on the outlier cluster (i.e., C2) drops significantly compared with that of FedAvg.

## 6 CONCLUSION

In this work, we provide an agent-level fairness measurement in FL (FAA) by taking agents' inherent heterogeneous data properties into account. Motivated by our fairness definition in FL, we also provide an effective FL training algorithm FOCUS to achieve high fairness. We theoretically analyze the convergence rate and optimality of FOCUS, and we prove that under mild conditions FOCUS is always fairer than the standard FedAvg protocol. We conduct thorough experiments on synthetic data with linear models as well as image and text datasets on deep neural networks. We show that FOCUS achieves stronger fairness than FedAvg and achieves similar or higher prediction accuracy across all datasets. We believe our work will inspire new research efforts on exploring the suitable fairness measurements for FL under different requirements.

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

## A  REVISION UPDATES

If the submission gets accepted, we will exploit the extra page for the main text to add the following contents.

### A.1  SCALABILITY WITH MORE AGENTS

To study the scalability of FOCUS, we evaluate the performance and fairness of FOCUS and existing methods under 100 clients on MNIST. Table 3 shows that FOCUS achieves the best fairness measured by FAA and Agnostic Loss, higher test accuracy, and lower test loss than Fedavg and existing fair FL methods.

Table 3: Comparison of different methods on MNIST 100 clients setting, in terms of average test accuracy (Avg Acc), average test loss (Avg Loss), fairness FAA and existing fairness metric Agnostic loss. FOCUS achieves the best fairness measured by FAA.

|  |  | FOCUS | FedAvg | q-FFL | AFL | Ditto | CGSV |
|---|---|---|---|---|---|---|---|
|  |  |  |  | $q = 1$ | $\lambda = 0.01$ | $\lambda = 1$ | $\beta = 1$ |
| Rotated MNIST (100 clients) | Avg Acc | **0.9533** | 0.9236 | 0.8371 | 0.8813 | 0.9351 | 0.8691 |
|  | Avg Loss | **0.157** | 0.2571 | 0.5668 | 0.4355 | 0.2206 | 0.6294 |
|  | FAA | **0.5605** | 1.0652 | 1.5055 | 0.8901 | 0.7459 | 1.2935 |
|  | Agnostic Loss | **0.5028** | 0.8894 | 1.4227 | 0.7767 | 0.620 | 1.5133 |

### A.2  CONVERGENCE OF FOCUS

We report the test accuracy and test loss of different methods over FL communication rounds on Rotated MNIST with 10/100 clients and Rotated CIFAR in Fig. 2. The results show that FOCUS converges faster and achieves higher accuracy and lower loss than other methods on both settings.

### A.3  RUNTIME ANALYSIS

**Computation time analysis for proposed metric FAA and its scalability to more clients.** In FAA, to calculate the maximal difference of excess risks for any pair of agents, it suffices to calculate the difference between the maximal per-client excess risks and the minimum per-client excess risk, and we don't need to calculate the difference for any pairs of agents. We compare the computation time (averaged over 100 trials) of FAA and existing fairness criteria (i.e., Accuracy Parity (Li et al., 2020b) and Agnostic Loss (Mohri et al., 2019) ) under 10 clients and 100 clients on MNIST. Table 4

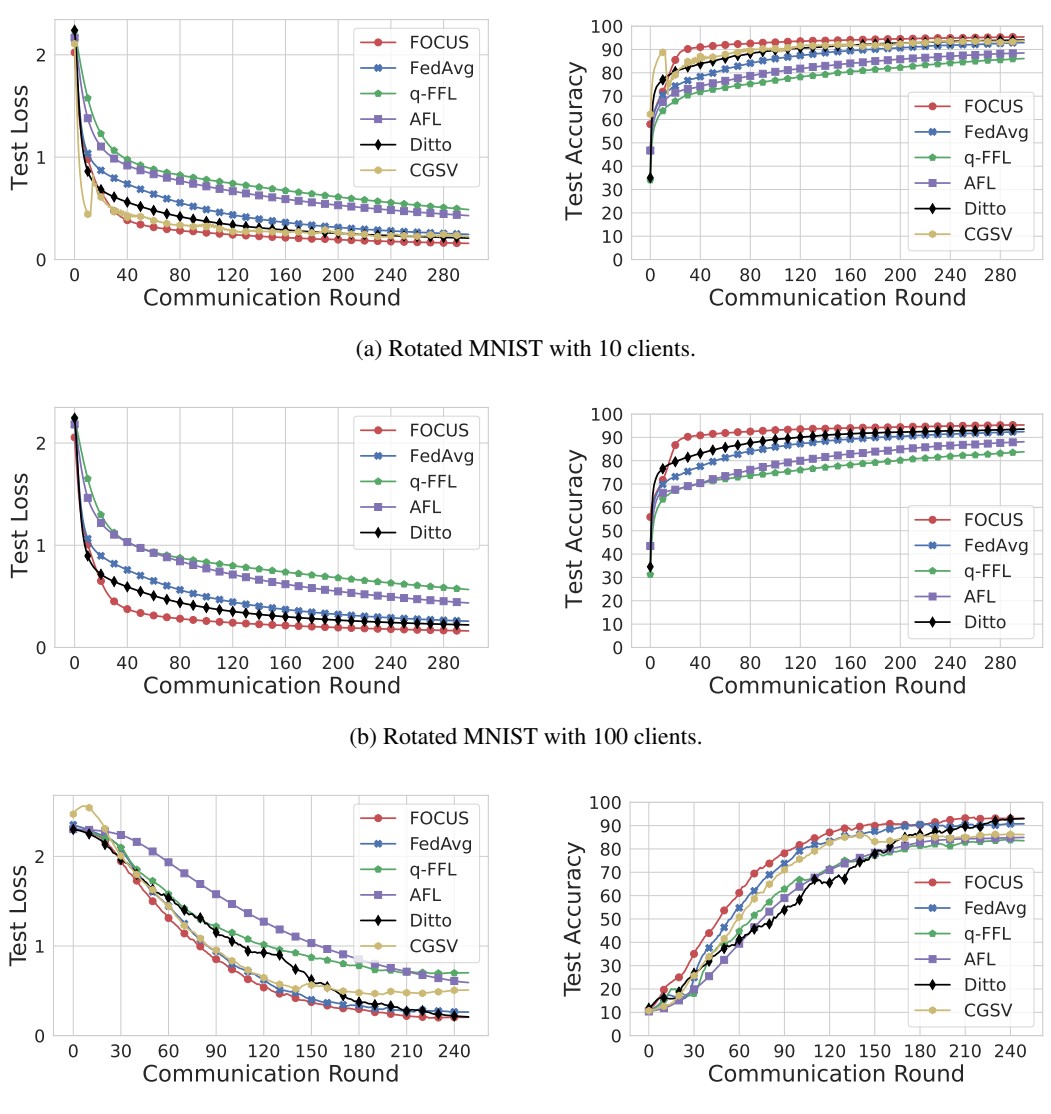

(a) Rotated MNIST with 10 clients.

(b) Rotated MNIST with 100 clients.

(c) Rotated CIFAR with 10 clients.

Figure 2: The test accuracy and test loss of different methods over FL communication rounds on different datasets. FOCUS converges faster and achieves higher accuracy and lower loss than other methods.

Table 4: Computation time of different FL fairness metric on Rotated MNIST. The computation of FAA is efficient under 10 clients and 100 client settings.

|  | 10 clients | 100 clients |
| --- | --- | --- |
| Accuracy Parity (Li et al., 2020b) | 4.70 e-05 second | 6.48 e-05 second |
| Agnostic loss (Mohri et al., 2019) | 9.41 e-07 second | 3.92 e-06 second |
| FAA | 6.09 e-06 second | 4.27 e-05 second |

shows that the computation of FAA is efficient even with a large number of agents. Moreover, calculating the difference between maximal excess risk and minimum excess risk (i.e., FAA) is even faster than calculating the standard deviation of the accuracy between agents (i.e., Accuracy Parity).

Table 5: The number of communication rounds that different methods take to reach a target accuracy on Rotated MNIST. FOCUS requires a significantly smaller number of communication rounds than other methods.

|        | 70% | 80% | 85% | 90%   |
|--------|-----|-----|-----|-------|
| FOCUS  | **9**  | **16**  | **20**  | **29**   |
| FedAvg | 10  | 51  | 88  | 177   |
| q-FFL  | 28  | 151 | 261 | > 300 |
| AFL    | 16  | 94  | 180 | > 300 |

Table 6: The number of communication rounds that different methods take to reach a target accuracy on Rotated CIFAR. FOCUS requires a significantly smaller number of communication rounds than other methods.

|        | 70% | 80% | 85%   | 90%   |
|--------|-----|-----|-------|-------|
| FOCUS  | **69**  | **92**  | **112**   | **135**   |
| FedAvg | 81  | 103 | 129   | 164   |
| q-FFL  | 102 | 159 | > 250 | > 250 |
| AFL    | 116 | 159 | 241   | > 250 |

**Communication rounds analysis.** Here, we report the number of communication rounds that each method takes to achieve targeted accuracy on MNIST and CIFAR in Table 5. We note that FOCUS requires significantly a smaller number of communication rounds than FedAvg, q-FFL, and AFL on both datasets, which demonstrates the small costs required by FOCUS.

**Training time and inference time analysis.** In terms of runtime, we report the training time for one FL round (averaged over 20 trials) as well as inference time (averaged over 100 trials) in Table 7. Since the local updates and sever aggregation for different cluster models can be run in parallel, we find that FOCUS has a similar training time compared to FedAvg, q-FFL, and AFL which train one global FL model. For the inference time, FOCUS is slightly slower than existing methods by about 0.17 seconds due to the ensemble prediction of all cluster models at each client. However, we note that such cost is small and the forward passes of different cluster models for the ensemble prediction can also be made in parallel to further reduce the inference time.

### A.4 COMPARISON TO FEDAVG WITH CLUSTERING

In this section, we construct a new method by combining the clustering and Fedavg together (i.e., FedAvg-HardCluster), which serves as a strong baseline. Specifically, FedAvg-HardCluster works as below:

- Step 1: before training, for each agent, it takes the arg max of the learned soft cluster assignment from FOCUS to get the hard cluster assignment (i.e., each agent only belongs to one cluster).

Table 7: Training time per FL round and inference time for different methods on Rotate MNIST.

|        | Training time per FL round | Inference time |
|--------|----------------------------|----------------|
| FOCUS  | 6.59 second                | 0.28 second    |
| FedAvg | 6.23 second                | 0.12 second    |
| q-FFL  | 6.32 second                | 0.11 second    |
| AFL    | 6.24 second                | 0.12 second    |

Table 8: Comparison between FOCUS and FedAvg-HardCluster on Rotate MNIST under two scenarios.

|  | Scenario 1 (underly clusters are clearly separatable) | | Scenario 2 (underly clusters are not separatable) | |
|---|---|---|---|---|
|  | FOCUS | FedAvg-HardCluster | FOCUS | FedAvg-HardCluster |
| Avg test acc | 0.953 | 0.954 | **0.814** | 0.812 |
| Avg test loss | 0.152 | 0.152 | **1.168** | 1.244 |
| FAA | **0.094** | 0.099 | **0.449** | 0.459 |
| Agnostic loss | 0.224 | 0.224 | **1.333** | 1.397 |

- Step 2: during training, each cluster then trains a FedAvg model based on corresponding agents.
- Step 3: during inference, each agent only uses the corresponding one cluster FedAvg model for inference.

To compare the performance between FOCUS and FedAvg-HardCluster, we consider two scenarios on MNIST:

- **Scenario 1**: underly clusters are clearly separatable, where each cluster contains samples from one distribution, which is the setting used in our paper.
- **Scenario 2**: underlying clusters are not separatable, where each cluster has 80%, 10%, and 10% samples from three different distributions, respectively. For example, the first underlying cluster contains 80% samples without rotation, 10% samples rotating 90 degrees, and 10% samples rotating 180 degrees.

We observe that the learned soft cluster assignments from FOCUS align with the underlying distribution, so the hard cluster assignment for Step 1 in FedAvg-HardCluster is equal to the underlying ground-truth clustering for both scenarios.

Table 8 presents the results of FOCUS and FedAvg-HardCluster on Rotated MNIST under two scenarios. **Under Scenario 1**, the accuracy of FOCUS and FedAvg-HardCluster is similar, and FOCUS achieves better fairness in terms of FAA. The results show that the hard clustering for FedAvg-HardCluster is as good as the soft clustering for FOCUS when the underlying clusters are clearly separable, which verifies that clustering is one of the key steps in FOCUS, and it aligns with our hypothesis for fairness under heterogeneous data. **Under Scenario 2**, FOCUS achieves higher accuracy and better FAA fairness than FedAvg-HardCluster. The results show that when underly clusters are not separatable, soft clustering is better than hard clustering since each agent can benefit from multiple cluster models with the soft $\pi$ learned from the EM algorithm in FOCUS.

## A.5 EFFECT OF THE NUMBER OF THE CLUSTERS $M$

The performance of FOCUS would not be harmed if the selected number of clusters is larger than the number of underlying clusters since the superfluous clusters would be useless (the corresponding soft cluster assignment $\pi$ goes to zero). On the other hand, when the selected number of clusters is smaller than the number of underlying clusters, FOCUS would converge to a solution when some clusters contain agents from more than one underlying cluster.

Empirically, in Table 9, we have 3 true underlying clusters while we set $M = 1, 2, 3, 4$ in our experiments, and we see that when $M = 3$ and $M = 4$, FOCUS achieves similar accuracy and fairness, which verifies our hypothesis that the superfluous clusters would become useless. When $M = 2$, FOCUS even achieves the highest fairness, which might be because one cluster benefits from the shared knowledge of multiple underlying clusters. When $M = 1$, FOCUS reduces to FedAvg, which does not have the clustering mechanism, leading to the lowest accuracy and fairness under heterogeneous data.

Table 9: The effect of $M$ on Rotate MNIST when the number of underlying clusters is 3.

|  | M=1 | M=2 | M=3 | M=4 |
|---|---|---|---|---|
| Avg test acc | 0.929 | 0.952 | **0.953** | **0.953** |
| Avg test loss | 0.246 | 0.167 | **0.152** | 0.153 |
| FAA | 0.363 | **0.079** | 0.094 | 0.091 |
| Agnostic loss | 0.616 | 0.272 | 0.224 | **0.223** |

# B  CONVERGENCE PROOF

## B.1  CONVERGENCE OF LINEAR MODELS (THEOREM 1)

### B.1.1  KEY LEMMAS

We need to state two lemmas first before proving Theorem 1.

**Lemma 1.** *Suppose $e \in S_m$ and the $m$-th cluster is the one closest to $w_m^*$. Assume $\|w_m^{(t)} - w_m^*\| \le \alpha < \beta \le \min_{m' \ne m} \|w_{m'}^{(t)} - w_m^*\|$. Then the E-step updates as*

$$\pi_{em}^{(t+1)} \ge \frac{\pi_{em}^{(t)}}{\pi_{em}^{(t)} + (1 - \pi_{em}^{(t)}) \exp\left(-(\beta^2 - \alpha^2 - 2(\alpha + \beta)r)\delta^2\right)} \tag{25}$$

**Remark.** Our assumption of proper initialization guarantees that $\|w_m^{(0)} - w_m^*\| \le \alpha$ while $\forall m'$, we have $\|w_{m'} - w_m^*\|_2 \ge \|w_m^* - \mu_{m'}^*\| - \|w_{m'} - \mu_{m'}^*\| = R - \alpha$. Hence, we substitute $\beta = R - \alpha$ and $\alpha = \frac{R}{2} - r - \Delta$, which yields

$$\pi_{em}^{(t+1)} \ge \frac{\pi_{em}^{(t)}}{\pi_{em}^{(t)} + (1 - \pi_{em}^{(t)}) \exp(-2R\Delta\delta^2)}, \quad \forall e \in S_m \tag{26}$$

For M-steps, the local agents are initialized with $\theta_{em}^{(0)} = w_m^{(t)}$. Then for $k = 1, \ldots, K - 1$, each agent use local SGD to update its personal model:

$$\theta_{em}^{(k+1)} = \theta_{em} - \eta_k g_{em}(\theta_{em}) = \theta_{em}^{(k)} - \eta_k \nabla \sum_{i=1}^{n_e} \ell(h_{\theta_{em}}(x_e^{(i)}), y_e^{(i)}). \tag{27}$$

To analyze the aggregated model Eq. (6), we define a sequence of virtual aggregated models $\hat{w}_m^{(k)}$.

$$\hat{w}_m^{(k)} = \sum_{e=1}^{E} \frac{\pi_{em} \theta_{em}^{(k)}}{\sum_{e'=1}^{E} \pi_{e'm}}. \tag{28}$$

**Lemma 2.** *Suppose any agent $e \in S_m$ has a soft clustering label $\pi_{em}^{(t+1)} \ge p$. Then one step of local SGD updates $\hat{w}_m^{(k)}$ by Eq. (29), if the learning rate $\eta_k \le \frac{1}{4\delta^2}$.*

$$\mathbb{E}\|\hat{w}_m^{(k+1)} - w_m^*\|_2^2 \le (1 - 2\eta_k \gamma_m p\delta^2)\mathbb{E}\|\hat{w}_m^{(k+1)} - w_m^*\|_2^2 + \eta_k A_1 + \eta_k^2 A_2. \tag{29}$$

$$A_1 = 4\gamma_m r\delta^2 + 2\delta^2 E(1 - p), \quad A_2 = 16E(K - 1)^2\delta^4 + O(\frac{d}{n_e})E(\delta^4 + \delta^2\sigma^2) \tag{30}$$

**Remark.** Using the recursive relation in Lemma 2, if the learning rate $\eta_k$ is fixed, the sequence $\hat{w}_m^{(k)}$ has a convergence rate of

$$\mathbb{E}\|\hat{w}_m^{(k)} - w_m^*\|_2^2 \le (1 - 2\eta\gamma_m p\delta^2)^k \mathbb{E}\|\hat{w}_m^{(0)} - w_m^*\|_2^2 + \eta k(A_1 + \eta A_2). \tag{31}$$

### B.1.2 COMPLETING THE PROOF OF THEOREM 1

We now combine Lemma 1 and Lemma 2 to prove Theorem 1. The theorem is restated below.

**Theorem 1.** *With the assumptions 1 and 2, $n_e = O(d)$, if learning rate $\eta \leq \min(\frac{1}{4\delta^2}, \frac{\beta}{\sqrt{T}})$,*

$$\pi_{em}^{(T)} \geq \frac{1}{1 + (M-1) \cdot \exp(-2R\delta^2 \Delta_0 K)}, \forall e \in S_m \tag{32}$$

$$\mathbb{E}\|w_m^{(T)} - w_m^*\|_2^2 \leq (1 - \frac{2\eta\gamma_m\delta^2}{M})^{KT}(\|w_m^{(0)} - w_m^*\|_2^2 + A) + 2MKr + \frac{M\delta^2 E\beta}{2\sqrt{T}}O(K^3, \sigma^2). \tag{33}$$

*where $K$ is the total number of communication rounds; $T$ is the number of iterations each round; $\gamma_m = |S_m|$ is the number of agents in the $m$-th cluster, and*

$$A = \frac{2EK(M-1)\delta^2}{(1 - \frac{2\eta\delta^2\gamma_m}{M})^K - \exp(-2R\delta^2\Delta_0)}. \tag{34}$$

*Proof.* We prove Theorem 1 by induction. Suppose

$$\pi_{em}^{(t)} \geq \frac{1}{1 + (M-1)\exp(-2R\delta^2\Delta_0 t)} \tag{35}$$

$$\mathbb{E}\|w_m^{(t)} - w_m^*\|^2 \leq (1 - \frac{2\eta\gamma_m\delta^2}{M})^{Kt}(\|w_m^{(0)} - w_m^*\|^2) + A\left((1 - \frac{2\eta\gamma_m\delta^2}{M})^{Kt} - \exp(-2R\delta^2\Delta_0 t)\right)$$
$$+ \frac{\eta B}{1 - (1 - \frac{2\eta\gamma_m\delta^2}{M})^K}. \tag{36}$$

where $B = [16E\delta^4 K^3 + EK(\delta^4 + \delta^2\sigma^2)]\eta + 4\gamma_m r\delta^2 K$.

Then according to Lemma 1,

$$\pi_{em}^{(t+1)} \geq \frac{\pi_{em}^{(t)}}{\pi_{em}^{(t)} + (1 - \pi_{em}^{(t)})\exp(-2R\Delta_0\delta^2)} \tag{37}$$

$$\geq \frac{1}{1 + (M-1)\exp(-2R\delta^2\Delta_0 t)\exp(-2R\Delta_0\delta^2)} \tag{38}$$

$$\geq \frac{1}{1 + (M-1)\exp(-2R\Delta_0\delta^2(t+1))}. \tag{39}$$

We recall the virtual sequence of $\hat{w}_m$ defined by Eq. (28). Since models are synchronized after $K$ rounds, the know $\hat{w}_m^{(0)} = w_m^{(t)}$ and $w_m^{(t+1)} = \hat{w}_m^{(K)}$. We then apply Lemma 2 to prove the induction. Note that instead of proving Eq. (33), we prove a stronger induction hypothesis of Eq. (36).

$$\mathbb{E}\|w_m^{(t+1)} - w_m^*\|^2$$
$$= \mathbb{E}\|\hat{w}_m^{(K)} - w_m^*\|^2 \tag{40}$$
$$\leq (1 - 2\eta\gamma_m p\delta^2)^K \mathbb{E}\|\hat{w}_m^{(t)} - w_m^*\|^2 + \eta K(A_1 + \eta A_2) \tag{41}$$
$$\leq (1 - 2\eta\gamma_m p\delta^2)^K \left((1 - \frac{2\eta\gamma_m\delta^2}{M})^{Kt}\|w_m^{(0)} - w_m^*\|^2 + A((1 - \frac{2\eta\gamma_m\delta^2}{M})^{Kt} - \exp(-2R\Delta_0\delta^2 t))\right.$$
$$\left. + \frac{\eta B}{1 - (1 - \frac{2\eta\gamma_m\delta^2}{M})^K}\right) + \eta K(4\gamma_m r\delta^2 + 2\delta^2 E(1-p)) + \eta^2 KA_2 \tag{42}$$
$$\leq (1 - \frac{2\eta\gamma_m\delta^2}{M})^{(t+1)K}\|w_m^{(0)} - w_m^*\|^2$$
$$+ \underbrace{A(1 - \frac{2\eta\gamma_m\delta^2}{M})^{(t+1)K} - A\exp(-2R\Delta_0\delta^2 t)(1 - \frac{2\eta\gamma_m\delta^2}{M})^K + 2\delta^2 E(1-p)}_{D_1}$$
$$+ \underbrace{(1 - \frac{2\eta\gamma_m\delta^2}{M})^K \frac{\eta B}{1 - (1 - \frac{2\eta\gamma_m\delta^2}{M})^K} + 4\eta K\gamma_m r\delta^2 + \eta^2 KA_2}_{D_2}. \tag{43}$$

Note that $1 - p \leq (M-1) \exp(-2R\Delta_0 \delta^2 t)$, so

$$D_1 \leq A(1 - \frac{2\eta\gamma_m\delta^2}{M})^{(t+1)K} - A\exp(-2R\Delta_0\delta^2 t)(1 - \frac{2\eta\gamma_m\delta^2}{M})^K + 2\delta^2 EK(M-1)\exp(-2R\Delta_0\delta^2 t)$$

$$\leq A((1 - \frac{2\eta\gamma_m\delta^2}{M})^{(t+1)K} - \exp(-2R\Delta_0\delta^2(t+1))) \tag{44}$$

For $D_2$ we have

$$D_2 \leq (1 - \frac{2\eta\gamma_m\delta^2}{M})^K \frac{\eta B}{[1 - (1 - \frac{2\eta\gamma_m\delta^2}{M})^K]} + 4\eta\gamma_m r\delta^2 K + 16\eta^2 E\delta^4 K^3 + \eta^2 EKO(\delta^4 + \delta^2\sigma^2)$$

$$= \frac{\eta B}{1 - (1 - \frac{2\eta\gamma_m\delta^2}{M})^K}. \tag{45}$$

Finally we combine Eqs. (43) to (45) so

$$\mathbb{E}\|w_m^{(t+1)} - w_m^*\|^2 \leq (1 - \frac{2\eta\gamma_m\delta^2}{M})^{(t+1)K}\|w_m^{(0)} - w_m^*\|^2 + A\left((1 - \frac{2\eta\gamma_m\delta^2}{M})^{(t+1)K} - \exp(-2R\delta^2\Delta_0(t+1))\right)$$

$$+ \frac{\eta B}{1 - (1 - \frac{2\eta\gamma_m\delta^2}{M})^K}. \tag{46}$$

Since it is trivial to check that both induction hypotheses hold when $t = 0$, the induction hypothesis holds. Note that $K \geq 1$, so

$$\frac{\eta B}{1 - (1 - \frac{2\eta\gamma_m\delta^2}{M})^K} \leq \eta B \frac{M}{2\eta\gamma_m\delta^2} \leq 2MKr + \frac{M\delta^2 E\beta}{2\sqrt{T}}O(K^3, \delta^2). \tag{47}$$

Combining Eq. (46) and Eq. (47) completes our proof. □

### B.1.3 Deferred Proofs of Key Lemmas

**Lemma 1.**

*Proof.* For simplicity, we abbreviate the model weights $w_m^{(t)}$ by $w_m$ in the proof of this lemma. The $n$-th E step updates the weights $\Pi$ by

$$\pi_{em}^{(t+1)} = \frac{\pi_{em}^{(t)} \exp\left[-\mathbb{E}_{(x,y)\sim D_e}(w_m^T x - y)^2\right]}{\sum_{m'} \pi_{em'}^{(t)} \exp\left[-\mathbb{E}_{(x,y)\sim D_e}(w_{m'}^T x - y)^2\right]} \tag{48}$$

so

$$\pi_{em}^{(t+1)} = \frac{\pi_{em}^{(t)} \exp\left(-\|w_m^{(t)} - \mu_e\|^2\delta^2\right)}{\sum_{m'} \pi_{em'}^{(t)} \exp\left[-\|w_m'^{(t)} - \mu_e\|^2\delta^2\right]} \tag{49}$$

$$\geq \frac{\pi_{em}^{(t)} \exp(-(\beta - r)^2\delta^2)}{\pi_{em}^{(t)} \exp(-(\beta - r)^2\delta^2) + \sum_{m'\neq m} \pi_{em'}^{(t)} \exp(-(\alpha + r)^2\delta^2)} \tag{50}$$

$$\geq \frac{\pi_{em}^{(t)}}{\pi_{em}^{(t)} + (1 - \pi_{em}^{(t)}) \exp\left(-(\beta^2 - \alpha^2 - 2(\alpha + \beta)r)\delta^2\right)} \tag{51}$$

□

**Lemma 2.**

*Proof.* Notice that local datasets are generated by $X_e \sim \mathcal{N}(0, \delta^2 \mathbf{1}^{n_e \times d})$ and $y_e = X_e\mu_e + \epsilon_e$ with $\epsilon_e \sim \mathcal{N}(0, \sigma^2)$. Therefore,

$$\|\hat{w}_m^{(k+1)} - w_m^*\|^2 = \|w_m^{(k)} - w_m^* - \eta_k g_k\|^2 \tag{52}$$

$$= \|\hat{w}_m^{(k)} - w_m^* - \eta_k \frac{2}{n_e} \sum_e \pi_{em} X_e^T X_e (\theta_{em}^{(k)} - \mu_e) + \frac{2\eta_k}{n_e} \sum_e \pi_{em} X_e^T \epsilon_e\|^2 \tag{53}$$

$$= \|\hat{w}_m^{(k)} - w_m^* - \hat{g}_k\|^2 + \eta_k^2 \|g_k - \hat{g}_k\|^2 + 2\eta_k \langle w_m^{(k)} - w_m^* - \hat{g}_k, \hat{g}_k - g_k \rangle. \tag{54}$$

where $\hat{g}_k = \frac{2}{n_e} \sum_e \pi_{em} \mathbb{E}(X_e^T X_e)(\theta_{em}^{(k)} - \mu)$. Since the expectation of the last term in Eq. (54) is zero, we only need to estimate the expectation of $\|\hat{w}_m^{(k)} - w_m^* - \eta_k \hat{g}_k\|^2$ and $\|\hat{g}_k - g_k\|^2$.

$$\|\hat{w}_m^{(k)} - w_m^* - \eta_k \hat{g}_k\|^2$$

$$= \|\hat{w}_m^{(k)} - w_m^*\|^2 + \frac{4\eta_k^2}{n_e^2} \sum_e \pi_{em} \mathbb{E}(X_e^T X_e) \|\theta_{em}^t - \mu_e\|^2 - \frac{4\eta_k}{n_e} \sum_e \pi_{em} \langle \hat{w}_m^{(k)} - w_m^*, \mathbb{E}(X_e^T X_e)(\theta_{em}^{(k)} - \mu_e) \rangle$$

$$= \|\hat{w}_m^{(k)} - w_m^*\|^2 + 4\eta_k^2 \delta^2 \sum_e \pi_{em} \|\theta_{em}^{(k)} - \mu_e\|^2 - \underbrace{4\eta_k \langle \hat{w}_m^{(k)} - w_m^*, \sum_e \pi_{em} \delta^2 (\theta_{em}^{(k)} - \mu_e) \rangle}_{C_1}. \tag{55}$$

$$C_1 = -4\eta_k \sum_e \pi_{em} \langle \hat{w}_m^{(k)} - \theta_{em}^{(k)}, \delta^2 (\theta_{em}^{(k)} - \mu_e) \rangle - 4\eta_k \sum_e \pi_{em} \langle \theta_{em}^{(k)} - w_m^*, \delta^2 (\theta_{em}^{(k)} - \mu_e) \rangle \tag{56}$$

$$\leq 4 \sum_e \pi_{em} \|\hat{w}_m^{(k)} - \theta_{em}^{(k)}\|^2 + 4\delta^4 \eta_k^2 \sum_e \pi_{em} \|\theta_{em}^{(k)} - \mu_e\|^2 - 4\eta_k \delta^2 \sum_e \pi_{em} \|\theta_{em}^{(k)} - \mu_e\|^2$$

$$- 4\eta_k \delta^2 \underbrace{\sum_e \pi_{em} \langle \mu_e - w_m^*, \theta_{em}^{(k)} - \mu_e \rangle}_{C_2} \tag{57}$$

Since $\eta_k \leq \frac{1}{4\delta^2}$,

$$\mathbb{E}\|\hat{w}_m^{(k)} - w_m^* - \eta_k \hat{g}_k\|^2 \tag{58}$$

$$\leq \mathbb{E}\|\hat{w}_m^{(k)} - w_m^*\|^2 + (8\delta^4 \eta_k^2 - 4\eta_k \delta^2) \sum_e \pi_{em} \mathbb{E}\|\theta_{em}^{(k)} - \mu_e\|^2 + 4 \sum_e \pi_{em} \mathbb{E}\|\hat{w}_m^{(k)} - \theta_{em}^{(k)}\|^2 + C_2 \tag{59}$$

$$\leq \mathbb{E}\|\hat{w}_m^{(k)} - w_m^*\|^2 - 2\eta_k \delta^2 \sum_e \pi_{em} \mathbb{E}\|\theta_{em}^{(k)} - \mu_e\|^2 + 4 \sum_e \pi_{em} \mathbb{E}\|\hat{w}_m^{(k)} - \theta_{em}^{(k)}\|^2 + C_2 \tag{60}$$

Note that

$$\sum_e \pi_{em} \mathbb{E}\|\theta_{em}^{(k)} - \mu_e\|^2 \tag{61}$$

$$= \sum_{e \in S_m} \pi_{em} \mathbb{E}\|\theta_{em}^{(k)} - \mu_e\|^2 + \sum_{e \notin S_m} \pi_{em} \mathbb{E}\|\theta_{em}^{(k)} - \mu_e\|^2 \tag{62}$$

$$\geq \sum_{e \in S_m} \pi_{em} (\mathbb{E}\|\theta_{em}^{(k)} - w_m^*\|^2 + 2r + r^2) + \sum_{e \notin S_m} \pi_{em} \mathbb{E}\|\theta_{em}^{(k)} - \mu_e\|^2 \tag{63}$$

$$= \sum_{e \in S_m} \pi_{em} (\mathbb{E}\|\hat{w}_m^{(k)} - w_m^*\|^2 + \mathbb{E}\|\hat{w}_m^{(k)} - \theta_{em}^{(k)}\|^2 + 2r + r^2) + \sum_{e \notin S_m} \pi_{em} \mathbb{E}\|\theta_{em}^{(k)} - \mu_e\|^2 \tag{64}$$

And since $\hat{w}_m^{(k)} = \mathbb{E}\sum_e \pi_{em}\theta_{em}^{(k)}$, we have

$$4\mathbb{E}\sum_e \pi_{em}\|\hat{w}_m^{(k)} - \theta_{em}^{(k)}\|^2 \leq 4\mathbb{E}\sum_e \pi_{em}\|\hat{w}_m^{(0)} - \theta_{em}^{(k)}\|^2 \tag{65}$$

$$\leq 4\sum_e \pi_{em}(K-1)\mathbb{E}\sum_{t'}^{t-1} {\eta_k'}^2\|\frac{2}{n_e}X_e^T X_e(\theta_{em}^{(k)} - \mu_e)\|^2 \tag{66}$$

$$\leq 16\eta_k^2 E(K-1)^2\delta^4. \tag{67}$$

Thus,

$$\mathbb{E}\|\hat{w}_m^{(k)} - w_m^* - \eta_k\hat{g}_k\|^2 \leq (1 - 2\eta_k\delta^2\sum_e \pi_{em})\mathbb{E}\|\hat{w}_m^{(k)} - w_m^*\|^2 + 16\eta_k^2 E(K-1)^2\delta^4$$

$$\underbrace{-2\eta_k\delta^2\sum_{e\notin S_m}\pi_{em}\mathbb{E}\|\theta_{em}^{(k)} - \mu_e\|^2 - 4\eta_k\delta^2\sum_e \pi_{em}\langle\theta_{em}^{(k)} - \mu_e, \mu_e - w_m^*\rangle}_{C_3}$$

$$\tag{68}$$

Since

$$C_3 \leq 2\eta_k\delta^2\sum_{e\notin S_m}\pi_{em}\|\mu_e - w_m^*\|_2^2 - 4\eta_k\delta^2\sum_{e\in S_m}\pi_{em}\|\theta_{em}^{(k)} - \mu_e\|_2\|\mu_e - w_m^*\|_2 \tag{69}$$

$$\leq 2\eta_k\delta^2 E(1-p) + 4\eta_k\delta^2\gamma_m r \tag{70}$$

we have

$$\mathbb{E}\|\hat{w}_m^{(k)} - w_m^* - \eta_k\hat{g}_k\|^2 \leq (2\eta_k\delta^2\gamma_m p)\mathbb{E}\|\hat{w}_m^{(k)} - w_m^*\|^2 + 16\eta_k^2 E(K-1)^2\delta^4 + 2\eta_k\delta^2 E(1-p) + 4\eta_k\delta^2\gamma_m r \tag{71}$$

Notice that

$$\mathbb{E}\|\hat{g}_k - g_k\|^2 = \mathbb{E}\sum_e \frac{4}{n_e^2}\pi_{em}\|(X_e^T X_e - \mathbb{E}(X_e^T X_e))(\theta_{em}^{(k)} - \mu_e)\|^2 + \mathbb{E}\sum_e \frac{4}{n_e^2}\sum_e \pi_{em}\|X_e^T \epsilon_e\|^2$$

$$= E\frac{O(dn_e)}{n_e^2}\delta^4 + E\frac{O(dn_e)}{n_e^2}\delta^2\sigma^2 \tag{72}$$

so

$$\mathbb{E}\|\hat{w}_m^{(k+1)} - w_m^*\|_2^2 \leq (1 - 2\eta_k\gamma_m p\delta^2)\mathbb{E}\|\hat{w}_m^{(k)} - w_m^*\|_2^2 + \eta_k A_1 + \eta_k^2 A_2 \tag{73}$$

where

$$A_1 = 4\delta^2\gamma_m r + 2\delta^2 E(1-p) \tag{74}$$

and

$$A_2 = 16E(K-1)^2\delta^4 + O(\frac{d}{n_e})E(\delta^4 + \delta^2\sigma^2). \tag{75}$$

$\square$

## B.2 CONVERGENCE OF MODELS WITH SMOOTH AND STRONGLY CONVEX LOSSES (THEOREM 2)

Here we present the detailed proof for Theorem 2.

### B.2.1 KEY LEMMAS

We first state two lemmas for E-step updates and M-step updates, respectively. The proofs of both lemmas are deferred to the Appendix B.2.3

**Lemma 3.** *Suppose the loss function $\mathcal{L}_{P_t}(\theta)$ is $L$-smooth and $\mu$-strongly convex for any cluster $m$. If $\|w_m^{(t)} - w_m^*\| \le \frac{\sqrt{\mu}R}{\sqrt{\mu}+\sqrt{L}} - r - \Delta$ for some $\Delta > 0$, then E-step updates as*

$$\pi_{em}^{(t)} \ge \frac{\pi_{em}^{(t)}}{\pi_{em}^{(t)} + (1 - \pi_{em}^{(t)})\exp(-\mu R\Delta)}. \tag{76}$$

For M-steps, the local agents are initialized with $\theta_{em}^{(0)} = w_m^{(t)}$. Then for $k = 1, \ldots, K-1$, each agent use local SGD to update its personal model:

$$\theta_{em}^{(k+1)} = \theta_{em} - \eta_k g_{em}(\theta_{em}) = \theta_{em}^{(k)} - \eta_k \nabla \sum_{i=1}^{n_e} \ell(h_{\theta_{em}}(x_e^{(i)}), y_e^{(i)}). \tag{77}$$

To analyze the aggregated model Eq. (6), we define a sequence of virtual aggregated models $\hat{w}_m^{(k)}$.

$$\hat{w}_m^{(k)} = \sum_{e=1}^{E} \frac{\pi_{em}\theta_{em}^{(k)}}{\sum_{e'=1}^{E} \pi_{e'm}}. \tag{78}$$

**Lemma 4.** *Suppose for any agent $e \in S_m$, its soft clustering label $\pi_{em}^{(t+1)} \ge p$. Then one step local SGD updates $\hat{w}_m^{(k)}$ by Eq. (79), if the learning rate $\eta_k \le \frac{1}{2(\mu+L)}$.*

$$\mathbb{E}\|\hat{w}_m^{(k+1)} - w_m^*\|_2^2 \le (1 - \eta_k A_0)\mathbb{E}\|\hat{w}_m^{(k)} - w_m^*\|_2^2 + \eta_k A_1 + \eta_k^2 A_2. \tag{79}$$

*where*

$$A_0 = \frac{2\gamma_m p\mu L}{\mu + L} \tag{80}$$

$$A_1 = 2\gamma_m Lr\sqrt{\frac{2G}{\mu}} + \frac{G(1-p)E}{\mu}(4L + \frac{6}{\mu+L}) + O(r^2). \tag{81}$$

$$A_2 = \frac{4E(K-1)^2 GL^2}{\mu} + \frac{E\sigma^2}{n_e}. \tag{82}$$

**Remark.** Using this recursive relation, if the learning rate $\eta_k$ is fixed, the sequence $\hat{w}_m^{(k+1)}$ has a convergence rate of

$$\mathbb{E}\|\hat{w}_m^{(k)} - w_m^*\|^2 \le (1 - \eta A_0)^k \mathbb{E}\|\hat{w}_m^{(0)} - w_m^*\|^2 + \eta k(A_1 + \eta A_2). \tag{83}$$

### B.2.2 COMPLETING THE PROOF OF THEOREM 2

**Theorem 2.** *Suppose loss functions have bounded variance for gradients on local datasets, i.e., $\mathbb{E}_{(x,y)\sim\mathcal{D}_e}[\|\nabla\ell(x,y;\theta) - \nabla\mathcal{L}_e(\theta)\|_2^2] \le \sigma^2$. Assume population losses are bounded, i.e., $\mathcal{L}_e \in G, \forall e \in [E]$. With initialization from assumptions 3 and 4, if each agent chooses learning rate $\eta \le \min(\frac{1}{2(\mu+L)}, \frac{\beta}{\sqrt{T}})$, the weights $(\Pi, W)$ converges by*

$$\pi_{em}^{(T)} \ge \frac{1}{1 + (M-1)\exp(-\mu R\Delta_0 T)}, \ \forall e \in S_m \tag{84}$$

$$\mathbb{E}\|w_m^{(T)} - w_m^*\|_2^2 \le (1 - \eta A)^{KT}(\|w_m^{(0)} - w_m^*\|_2^2 + B) + O(Kr) + \frac{ME\beta O(K^3, \frac{\sigma^2}{n_e})}{\sqrt{T}} \tag{85}$$

*where $T$ is the total number of communication rounds; $K$ is the number of iterations each round; $\gamma_m = |S_m|$ is the number of agents in the m-th cluster, and*

$$A = \frac{2\gamma_m}{M}\frac{\mu L}{\mu + L}, B = \frac{GMTE(\frac{4L}{\mu} + \frac{6}{\mu(\mu+L)})}{(1 - \eta A)^K - \exp(-\mu R\Delta_0)}. \tag{86}$$

*Proof.* The proof is quite similar to Theorem 1 for linear models: we follow an induction proof using lemmas 3 and 4. Suppose Eq. (84) hold for step $t$. And suppose

$$\mathbb{E}\|w_m^{(t)} - w_m^*\|_2^2 \leq (1-\eta A)^{Kt}(\|w_m^{(0)} - w_m^*\|_2^2) + B((1-\eta A)^{Kt} - \exp(-\mu R\Delta_0 t)) + \frac{\eta C}{1-(1-\eta A)^K}. \quad (87)$$

where

$$C = \frac{4\eta EGK^3 L^2}{\mu} + (2\gamma_m Lr\sqrt{\frac{2G}{\mu}} + O(r^2)) + \eta\frac{EK\sigma^2}{n_e}. \quad (88)$$

Then for any $t \in S_m$,

$$\pi_{em}^{(t+1)} \geq \frac{\pi_{em}^{(t)}}{\pi_{em}^{(t)} + (1-\pi_{em}^{(t)})\exp(-\mu R\Delta_t)} \quad (89)$$

$$\geq \frac{1}{1 + (M-1)\exp(-\mu R\Delta_0 t)\exp(-\mu R\Delta_t)} \quad (90)$$

$$\geq \frac{1}{1 + (M-1)\exp(-\mu R\Delta_0(t+1))} \quad (91)$$

We recall the virtual sequence $\hat{w}_m^{(k)}$ defined in Eq. (78). Models are synchronized after $K$ rounds of local iterations, so $w_m^{(t+1)} = \hat{w}_m^{(K)}$. Thus, according to Lemma 4,

$$\mathbb{E}\|w_m^{(t+1)} - w_m^*\|_2^2 = \mathbb{E}\|\hat{w}_m^{(K)} - w_m^*\|_2^2 \quad (92)$$

$$\leq (1-\eta A_0)^K \mathbb{E}\|w_m^{(t)} - w_m^*\|_2^2 + \eta K(A_1 + \eta A_2) \quad (93)$$

$$\leq (1-\eta A_0)^K \left((1-\eta A)^{Kt}(\mathbb{E}\|w_m^{(0)} - w_m^*\|^2) + B((1-\eta A)^{Kt} - \exp(-\mu R\Delta_0 t)) + \frac{\eta C}{1-(1-\eta A)^K}\right) + \eta K(A_1 + \eta A_2) \quad (94)$$

$$\leq (1-\eta A)^{(t+1)K}\mathbb{E}\|w_m^{(0)} - w_m^*\|^2 + \underbrace{(1-\eta A)^K B((1-\eta A)^{Kt} - \exp(-\mu R\Delta_0 t)) + \eta\frac{GK(1-p)E}{\mu}(4L + \frac{6}{\mu+L})}_{F_1}$$

$$+ \underbrace{(1-\eta A)^K \frac{\eta C}{1-(1-\eta A)^K} + \eta K(2\gamma_m Lr\sqrt{\frac{2G}{\mu}} + O(r^2)) + \eta^2 K A_2}_{F_2}. \quad (95)$$

For $F_1$, we use the fact that

$$\pi_{em}^{(t+1)} \geq \frac{1}{1 + (M-1)\exp-(\mu R\Delta_0(t+1))} \geq 1 - (M-1)\exp(-\mu R\Delta(t+1)),$$

so

$$F_1 \leq (1-\eta A)^K B((1-\eta A)^{Kt} - \exp(-\mu R\Delta_0 t)) + \eta\frac{G(M-1)\exp(-\mu R\Delta_0 t)}{\mu}(4L + \frac{6}{\mu+L}) \quad (96)$$

$$= B\left((1-\eta A)^{(t+1)K} - \exp(-\mu R\Delta_0 t)\right) \quad (97)$$

For $F_2$, we have

$$F_2 \leq (1-\eta A)^K \frac{\eta C}{1-(1-\eta A)^K} + \eta K(2\gamma_m Lr\sqrt{\frac{2G}{\mu}} + O(r^2)) + \frac{4EGL^2\eta^2 K^3}{\mu} + \frac{\eta^2 KE\sigma^2}{n_e} \quad (98)$$

$$\leq \frac{\eta C}{1-(1-\eta A)^K}. \quad (99)$$

Combining $F_1$ and $F_2$ finishes the induction proof. Moreover, since $T \geq 1$, we have

$$\frac{\eta C}{1-(1-\eta A)^K} \leq \frac{C}{A} = O(Kr) + \frac{ME\beta}{\sqrt{T}}O(K^3, \frac{\sigma^2}{n_e}). \quad (100)$$

Combining Eq. (87) and Eq. (100) completes our proof. □

### B.2.3 DEFERRED PROOFS OF KEY LEMMAS

**Lemma 3.**

*Proof.* According to Algorithm 1,

$$\pi_{em}^{(t+1)} = \frac{\pi_{em}^{(t)}}{\pi_{em}^{(t)} + \sum_{m' \neq m} \pi_{em'}^{(t)} \exp\left(\mathbb{E}\ell(x, y; w_m^{(t)}) - \mathbb{E}\ell(x, y; w_{m'}^{(t)})\right)} \tag{101}$$

$$\geq \frac{\pi_{em}^{(t)}}{\pi_{em}^{(t)} + (1 - \pi_{em}^{(t)}) \exp\left(\max_{m' \neq m}(\mathcal{L}_{P_e}(w_m^{(t)}) - \mathcal{L}_{P_e}(w_{m'}^{(t)}))\right)} \tag{102}$$

Since $\mathcal{L}_{P_e}$ is $L$-smooth and $\mu$-strongly convex,

$$\mathcal{L}_{P_e}(w_m^{(t)}) - \mathcal{L}_{P_e}(w_{m'}^{(t)}) \leq \frac{L}{2}\|w_m^{(t)} - \theta_t^*\|^2 - \frac{\mu}{2}\|w_{m'}^{(t)} - \theta_t^*\|^2$$

$$\leq \frac{L}{2}\left(\frac{\sqrt{\mu}R}{\sqrt{\mu} + \sqrt{L}} - \Delta\right)^2 - \frac{\mu}{2}\left(\frac{\sqrt{L}R}{\sqrt{\mu} + \sqrt{L}} + \Delta\right)^2$$

$$\leq -\sqrt{\mu L}R\Delta + \frac{L - \mu}{2}\Delta^2 \leq -\mu R\Delta. \tag{103}$$

Combining Eq. (102) and Eq. (103) completes our proof. □

**Lemma 4.**

*Proof.* We define $g_m^{(k)} = \sum_e \pi_{em} \frac{1}{n_e} \sum_{i=1}^{n_e} \nabla \ell(h_{\theta_{em}}(x_e^{(i)}), y_e^{(i)})$ and $\hat{g}_m^{(k)} = \sum_e \pi_{em} \nabla \mathcal{L}(\theta_{em}^{(k)})$.

$$\mathbb{E}\|\hat{w}_m^{(k+1)} - w_m^*\|^2 = \mathbb{E}\|\hat{w}_m^{(k)} - w_m^* - \eta_k g_m^{(k)}\|^2 \tag{104}$$

$$= \mathbb{E}\|\hat{w}_m^{(k)} - w_m^* - \eta_k \hat{g}_m^{(k)}\|^2 + \eta_k^2 \mathbb{E}\|g_m^{(k)} - \hat{g}_m^{(k)}\|^2$$

$$+ 2\eta_k \mathbb{E}\langle w_m^{(k)} - w_m^* - \eta_k \hat{g}_m^{(k)}, \hat{g}_m^{(k)} - g_m^{(k)}\rangle \tag{105}$$

$$= \mathbb{E}\|\hat{w}_m^{(k)} - w_m^* - \eta_k \hat{g}_m^{(k)}\|^2 + \eta_k^2 \mathbb{E}\|g_m^{(k)} - \hat{g}_m^{(k)}\|^2. \tag{106}$$

The first term can be decomposed into

$$\|\hat{w}_m^{(k)} - w_m^* - \eta_k \hat{g}_m^{(k)}\|^2 = \|\hat{w}_m^{(k)} - w_m^*\|^2 + \eta_k^2\|\hat{g}_m^{(k)}\|^2 - 2\eta_k\langle \hat{w}_m^{(k)} - w_m^*, \hat{g}_m^{(k)}\rangle. \tag{107}$$

Note that

$$\|\hat{g}_m^{(k)}\|^2 \leq \sum_{e=1}^{E} \pi_{em}\|\nabla\mathcal{L}_e(\theta_{em}^{(k)})\|^2. \tag{108}$$

$$-\langle \hat{w}_m^{(k)} - w_m^*, \hat{g}_m^{(k)}\rangle = -\sum_{e=1}^{E} \pi_{em}\langle \hat{w}_m^{(k)} - \theta_{em}^{(k)}, \nabla\mathcal{L}_e(\theta_{em}^{(k)})\rangle - \sum_{e=1}^{E} \pi_{em}\langle \theta_{em}^{(k)} - w_m^*, \nabla\mathcal{L}_e(\theta_{em}^{(k)})\rangle. \tag{109}$$

We further decompose the two terms in Eq. (109) by

$$-2\langle \hat{w}_m^{(k)} - \theta_{em}^{(k)}, \nabla\mathcal{L}_e(\theta_{em}^{(k)})\rangle \leq \frac{1}{\eta_k}\|\hat{w}_m^{(k)} - \theta_{em}^{(k)}\|^2 + \eta_k\|\nabla\mathcal{L}_e(\theta_{em}^{(k)})\|^2. \tag{110}$$

and

$$\langle \theta_{em}^{(k)} - w_m^*, \nabla\mathcal{L}_e(\theta_{em}^{(k)})\rangle \geq \langle \theta_{em}^{(k)} - w_m^*, \nabla\mathcal{L}_e(\theta_{em}^{(k)}) - \nabla\mathcal{L}_e(w_m^*)\rangle + \|\nabla\mathcal{L}_e(w_m^*)\|_2\|\theta_{em}^{(k)} - w_m^*\|_2. \tag{111}$$

$$\geq \frac{\mu L}{\mu + L}\|\theta_{em}^{(k)} - w_m^*\|^2 + \frac{1}{\mu + L}\|\nabla\mathcal{L}_e(\theta_{em}^{(k)} - \nabla\mathcal{L}_e(w_m^*))\|^2 + \|\nabla\mathcal{L}_e(w_m^*)\|_2\|\theta_{em}^{(k)} - w_m^*\|_2. \tag{112}$$

Therefore,

$$\mathbb{E}\|\hat{w}_m^{(k+1)} - w_m^*\|^2 = \underbrace{\mathbb{E}\|\hat{w}_m^{(k)} - w_m^*\|^2 - 2\eta_k \frac{\mu L}{\mu+L}\sum_e \pi_{em}\mathbb{E}\|\theta_{em}^{(k)} - w_m^*\|^2}_{E_1} + \underbrace{\sum_e \pi_{em}\mathbb{E}\|\hat{w}_m^{(k)} - \theta_{em}^{(k)}\|^2}_{E_2}$$

$$+ \underbrace{\left(2\eta_k^2 \sum_e \pi_{em}\mathbb{E}\|\nabla\mathcal{L}_e(\theta_{em}^{(k)})\|^2 - 2\eta_k \frac{1}{\mu+L}\sum_e \pi_{em}\mathbb{E}\|\nabla\mathcal{L}_e(\theta_{em}^{(k)}) - \nabla\mathcal{L}_e(w_m^*)\|^2\right)}_{E_3}$$

$$+ \underbrace{2\eta_k \mathbb{E}\sum_e \pi_{em}\|\theta_{em}^{(k)} - w_m^*\|_2 \cdot \|\nabla\mathcal{L}_e(w_m^*)\|_2}_{E_4} + \underbrace{\eta_k^2\mathbb{E}\|g_m^{(k)} - \hat{g}_m^{(k)}\|^2}_{E_5}. \tag{113}$$

$$\square$$

$$E_1 = \mathbb{E}\|\hat{w}_m^{(k)} - w_m^*\|^2 - 2\eta_k \frac{\mu L}{\mu+L}\mathbb{E}\left(\sum_e \pi_{em}\|\hat{w}_m^{(k)} - w_m^*\|^2 + \sum_e \pi_{em}\|\hat{w}_m^{(k)} - \theta_{em}^{(k)}\|^2\right)$$

$$\leq (1 - \frac{2\eta_k\mu L p\gamma_m}{\mu+L})\mathbb{E}\|w_m^{(k)} - w_m^*\|^2 + E_2. \tag{114}$$

$$E_2 = \mathbb{E}\sum_e \pi_{em}\|\hat{w}_m^{(k)} - \theta_{em}^{(k)}\|^2$$

$$= \mathbb{E}\sum_e \pi_{em}\|(w_m^{(0)} - \theta_{em}^{(k)}) + (\theta_{em}^{(k)} - w_m^{(k)})\|^2$$

$$\leq \mathbb{E}\sum_e \pi_{em}\|(w_m^{(0)} - \theta_{em}^{(k)})\|^2$$

$$\leq \sum_e \pi_{em}(K-1)\mathbb{E}\sum_{k'=0}^{k-1}\eta_{k'}^2\|g_{em}(\theta_{em}^{(k')})\|^2$$

$$\leq \frac{2\eta_k^2 E(K-1)^2 G^2 L^2}{\mu}. \tag{115}$$

$$E_3 = 2\mathbb{E}\sum_e \pi_{em}\left((\eta_k^2 - \frac{\eta_k}{\mu+L})\|\nabla\mathcal{L}_e(\theta_{em}^{(k)})\|^2 + \frac{2\eta_k}{\mu+L}\langle\nabla\mathcal{L}_e(\theta_{em}^{(k)}), \nabla\mathcal{L}_e(w_m^*)\rangle - \eta_k\frac{\|\nabla\mathcal{L}_e(w_m^*)\|^2}{\mu+L}\right)$$

$$\leq 2\eta_k\mathbb{E}\sum_e \pi_{em}\left(\frac{1}{2(\mu+L)}\|\nabla\mathcal{L}_e(\theta_{em}^{(k)})\|^2 + \frac{1}{\mu+L}\langle\nabla\mathcal{L}_e(\theta_{em}^{(k)}), \nabla\mathcal{L}_e(w_m^*)\rangle - \frac{\|\nabla\mathcal{L}_e(\theta_{em}^{(k)})\|^2}{\mu+L}\right)$$

$$\leq 6\eta_k\mathbb{E}\frac{\|\nabla\mathcal{L}_e(w_m^*)\|^2}{\mu+L}$$

$$\leq 6\eta_k \sum_{e\in S_m}\pi_{em}\frac{L^2 r^2}{\mu+L} + 6\eta_k \sum_{e\notin S_m}\pi_{em}\frac{2G}{\mu(\mu+L)}$$

$$\leq \eta_k O(r^2) + 6\eta_k\frac{G(1-p)E}{\mu(\mu+L)}. \tag{116}$$

$$E_4 = 2\eta_k\mathbb{E}\sum_{e\in S_m}\pi_{em}\|\theta_{em}^{(k)} - w_m^*\|_2 \cdot \|\nabla\mathcal{L}_e(w_m^*)\|_2 + 2\eta_k\mathbb{E}\sum_{e\notin S_m}\pi_{em}\|\theta_{em}^{(k)} - w_m^*\|_2 \cdot \|\nabla\mathcal{L}_e(w_m^*)\|_2$$

$$\leq 2\eta_k\gamma_m L r\sqrt{\frac{2G}{\mu}} + 2\eta_k(1-p)EL \cdot \frac{2G}{\mu}. \tag{117}$$

$$E_5 = \eta_k^2 \mathbb{E}\|g_m^{(k)} - \hat{g}_m^{(k)}\|^2$$

$$\leq \eta_k^2 \mathbb{E}\Big\|\sum_e \pi_{em}\Big(\frac{1}{n_e}\sum_{i=1}^{n_e}\nabla\ell(h_{\theta_{em}}(x_e^{(i)}), y_e^{(i)}) - \mathcal{L}(\theta_{em}^{(k)})\Big)\Big\|^2$$

$$\leq \eta_k^2 E\frac{\sigma^2}{n_e}. \tag{118}$$

Combining Eq. (114) to Eq. (118) yields the conclusion of Lemma 4.

## C  FAIRNESS ANALYSIS

### C.1  PROOF OF THEOREM 3

*Proof.* Let the first cluster $m_1$ contain agents $\mu_1, \ldots, \mu_{E-1}$, while the second cluster contains only the outlier $\mu_E$. Then, for $e = 1, \ldots, E-1$,

$$\mathcal{E}_e(w_{m_1}) = \delta^2\Big\|\mu_e - \frac{\sum_{e'=1}^{E-1}\mu_{e'}}{E-1}\Big\|^2 \leq \delta^2 r^2 \tag{119}$$

And for the outlier agent, the expected output is just the optimal solution, so

$$\mathcal{E}_E(w_{m_2}) = 0 \tag{120}$$

As a result, the fairness of this algorithm is bounded by

$$\mathcal{FAA}_{focus}(P) = \max_{i,j\in[E]}|\mathcal{E}_i(\Pi, W) - \mathcal{E}_j(\Pi, W)| \leq \delta^2 r^2. \tag{121}$$

On the other hand, the expected final weights of of FedAvg algorithm is $w_{avg} = \bar{\mu} = \frac{\sum_{e=1}^{E}\mu_e}{E}$, so the expected loss for agent $e$ shall be

$$\mathbb{E}_{(x,y)\sim\mathcal{P}_e}(\ell_{\hat{\theta}}(x)) = \mathbb{E}_{x\sim\mathcal{N}(0,\delta^2 I_d),\epsilon\sim\mathcal{N}(0,\sigma_e^2)}[(\mu_i^T x + \epsilon - \bar{\mu}^T x)^2] = \sigma_e^2 + \delta^2\|\mu_e - \bar{\mu}\|^2 \tag{122}$$

The infimum risk for agent $t_1$ is $\sigma_1^2$, and after subtracting it from the expected loss, we have

$$\mathcal{E}_1(w_{avg}) = \delta^2\|\mu_1 - \bar{\mu}\|^2 \tag{123}$$

$$= \delta^2\|\mu_1 - \frac{\sum_{e=1}^{E-1}\mu_1}{E} - \frac{\mu_E}{E}\|^2 \tag{124}$$

$$\leq \delta^2\Big(r \cdot \frac{E-1}{E} + \frac{\|\mu_1 - \mu_E\|}{E}\Big)^2 \tag{125}$$

$$\leq \delta^2\Big(r \cdot \frac{E-1}{E} + \frac{R+r}{E}\Big)^2 = \delta^2\Big(r + \frac{R}{E}\Big)^2 \tag{126}$$

However for the outlier agent,

$$\mathcal{E}_E(w_{avg}) = \delta^2\|\mu_E - \bar{\mu}\|^2 \tag{127}$$

$$= \delta^2\Big\|\frac{E-1}{E}\mu_E - \frac{\sum_{e=1}^{E-1}\mu_E}{E}\Big\|^2 \tag{128}$$

$$\geq \Big(\frac{E-1}{E}\Big)^2\delta^2 R^2 \tag{129}$$

Hence,

$$\mathcal{FAA}_{avg}(P) \geq \mathcal{E}_E(w_{avg}) - \mathcal{E}_1(w_{avg}) = \delta^2\Big(\frac{R^2(E-2) - 2Rr}{E} + r^2\Big) \tag{130}$$

$\square$

**Remark.** When there are $E_k > 1$ outliers, we can similarly derive FAA for FedAvg algorithm:

$$\mathcal{E}_1(w_{avg}) \leq \delta^2 (r + \frac{E_k R}{E})^2 \tag{131}$$

$$\mathcal{E}_E(w_{avg}) \geq \delta^2 (\frac{E - E_k}{E} R - \frac{E_k}{E} r)^2 \tag{132}$$

so as long as $E_k < \frac{E}{2}$,

$$\mathcal{FAA}_{avg} \geq \mathcal{E}_E(w_{avg}) - \mathcal{E}_1(w_{avg}) = \Omega(\delta^2 R^2) \tag{133}$$

The FOCUS algorithm produces a result with

$$\mathcal{E}_1(w_{m_1}) \leq \delta^2 r^2 \tag{134}$$

$$\mathcal{E}_E(w_{m_2}) \leq \delta^2 r^2 \tag{135}$$

Hence we still have

$$\mathcal{FAA}_{focus} \leq \delta^2 r^2. \tag{136}$$

## C.2 PROOF OF THEOREM 4

*Proof.* Note that the local population loss for agent $i$ with weights $\theta$ is

$$\mathcal{L}_i(\theta) = \int p_i(x, y) \ell(f_\theta(x), y) \mathrm{d}x \mathrm{d}y. \tag{137}$$

Thus,

$$|\mathcal{L}_i(\theta_i^*) - \mathcal{L}_j(\theta_i^*)| = \int |p_i(x, y) - p_j(x, y)| \cdot \ell(f_{\theta_i^*}(x), y) \mathrm{d}x \mathrm{d}y \tag{138}$$

$$\leq G \cdot \int |p_i(x, y) - p_j(x, y)| \mathrm{d}x \mathrm{d}y \leq Gr. \tag{139}$$

Hence,

$$\mathcal{L}_i(\theta_j^*) \leq \mathcal{L}_j(\theta_j^*) + Gr \leq \mathcal{L}_j(\theta_i^*) + Gr \leq \mathcal{L}_i(\theta_i^*) + 2Gr. \tag{140}$$

For the cluster that combines agents $\{1, \ldots, E - 1\}$ together, the weight converges to $\bar{\theta}' = \frac{1}{E-1} \sum_{i=1}^{E-1} \theta_i^*$. Then $\forall i = 1, \ldots, E - 1$, the population loss for the ensemble prediction

$$\mathcal{L}_i(\theta, \Pi) = \mathcal{L}_i \Big( \frac{\sum_{j=1}^{E-1} \theta_j^*}{E - 1} \Big) \tag{141}$$

$$\leq \frac{1}{T - 1} \sum_{j=1}^{T-1} \mathcal{L}_i(\theta_j^*) \tag{142}$$

$$\leq \mathcal{L}_i(\theta_i^*) + \frac{2Gr}{E - 1}. \tag{143}$$

Therefore, for any $i = 1, \ldots, T - 1$,

$$\mathcal{E}_i(\theta, \Pi) \leq \frac{2Gr}{E - 1}. \tag{144}$$

Since $\mathcal{E}_T(\theta, \Pi) = 0$,

$$\mathcal{FAA}_{focus}(W, \Pi) \leq \frac{2Gr}{E - 1} \tag{145}$$

Now we prove the second part of Theorem 4 for the fairness of Fedavg algorithm. For simplicity, we define $B = \frac{2Gr}{E-1}$ in this proof. Also, we denote the mean of all optimal weight $\bar{\theta} = \frac{\sum_{i=1}^{E} \theta_i^*}{E}$ and $\bar{\theta}' = \frac{\sum_{i=1}^{E-1} \theta_i^*}{E-1}$.

Remember that we assume loss functions to be L-smooth, so

$$\mathcal{L}_E(\theta_i^*) \le \mathcal{L}_E(\bar{\theta}') + \langle \nabla \mathcal{L}_E(\bar{\theta}'), \theta_i^* - \bar{\theta}' \rangle + \frac{L}{2} \|\bar{\theta}' - \theta_i\|^2. \tag{146}$$

Taking summation over $i = 1, \ldots, E - 1$, we get

$$\mathcal{L}_E(\bar{\theta}') \ge \frac{1}{E-1} \Big( \sum_{i=1}^{E-1} \mathcal{L}_E(\theta_i^*) - \langle \nabla \mathcal{L}_E(\bar{\theta}'), \sum_{i=1}^{E-1} (\theta_i - \bar{\theta}') \rangle - \frac{L}{2} \sum_{i=1}^{E-1} \|\bar{\theta}' - \theta_i\|^2 \Big) \tag{147}$$

$$= \frac{1}{E-1} \Big( \sum_{i=1}^{E-1} \mathcal{L}_E(\theta_i^*) - \frac{L}{2} \sum_{i=1}^{E-1} \|\bar{\theta}' - \theta_i\|^2 \Big) \tag{148}$$

$$\ge \mathcal{L}_E(\theta_E^*) + R - \frac{LB}{\mu}. \tag{149}$$

The last inequality uses the $\mu$-strongly convex condition that implies

$$B \ge \mathcal{L}_i(\bar{\theta}') - \mathcal{L}_i(\theta_i^*) \ge \frac{\mu}{2} \|\bar{\theta}' - \theta_i\|^2. \tag{150}$$

By $L$-smoothness, we have

$$\mathcal{L}_E(\bar{\theta}') \le \mathcal{L}_E(\bar{\theta}) + \langle \nabla \mathcal{L}_E(\bar{\theta}), \bar{\theta}' - \bar{\theta} \rangle + \frac{L}{2} \|\bar{\theta}' - \bar{\theta}\|^2. \tag{151}$$

$$\mathcal{L}_E(\theta_E^*) \le \mathcal{L}_E(\bar{\theta}) + \langle \nabla \mathcal{L}_E(\bar{\theta}), \theta_E^* - \bar{\theta} \rangle + \frac{L}{2} \|\theta_E^* - \bar{\theta}\|^2. \tag{152}$$

Note that $\bar{\theta} = \frac{\bar{\theta}' + (E-1)\theta_E^*}{E}$, we take a weighted sum over the above two inequalities to cancel the dot product terms out. We thus derive

$$\mathcal{L}_E(\bar{\theta}) \ge \frac{(E-1)\mathcal{L}_E(\bar{\theta}') + \mathcal{L}_E(\theta_E^*) - \frac{L}{2}(E-1)\|\bar{\theta}' - \bar{\theta}\|^2 - \frac{L}{2}\|\theta_E^* - \bar{\theta}\|^2}{E} \tag{153}$$

$$= \frac{E-1}{E} \Big( R - \frac{LB}{\mu} - \frac{L\|\theta_E^* - \bar{\theta}'\|^2}{2E} \Big) + \mathcal{L}_E(\theta_E^*). \tag{154}$$

Note that $\mathcal{L}_E(\cdot)$ is $\mu$-strongly convex, which means

$$R - \frac{LB}{\mu} \ge \mathcal{L}_E(\bar{\theta}') - \mathcal{L}_E(\theta_E^*) \ge \frac{\mu}{2} \|\theta_E^* - \bar{\theta}'\|^2. \tag{155}$$

so

$$\mathcal{L}_E(\bar{\theta}) \ge (1 - \frac{L}{\mu E}) \cdot \frac{E-1}{E} (R - \frac{LB}{\mu}) + \mathcal{L}_E(\theta_E^*). \tag{156}$$

And

$$\mathcal{E}_E(\bar{\theta}) \ge (1 - \frac{L}{\mu E}) \cdot \frac{E-1}{E} (R - \frac{LB}{\mu}). \tag{157}$$

On the other hand, for agent $i = 1, \ldots, E - 1$ we know

$$\mathcal{L}_i(\bar{\theta}) \le \mathcal{L}_i(\bar{\theta}') + \langle \nabla \mathcal{L}_i(\bar{\theta}'), \bar{\theta} - \bar{\theta}' \rangle + \frac{L}{2} \|\bar{\theta} - \bar{\theta}'\|^2. \tag{158}$$

By $L$ smoothness,

$$\|\nabla \mathcal{L}_i(\bar{\theta}')\|_2 \le L \|\bar{\theta}' - \theta_i^*\| \le L \sqrt{\frac{2B}{\mu}}. \tag{159}$$

So

$$\mathcal{L}_i(\bar{\theta}) \le \mathcal{L}_i(\theta_i^*) + B + L\sqrt{\frac{2B}{\mu}} \sqrt{\frac{2(R - \frac{LB}{\mu})}{\mu}} \frac{1}{E} + \frac{L(R - \frac{LB}{\mu})}{\mu E^2} \tag{160}$$

$$\mathcal{E}_i(\bar{\theta}) \leq B + \frac{2L}{\mu E}\sqrt{B(R - \frac{LB}{\mu})} + \frac{L(R - \frac{LB}{\mu})}{\mu E^2} \tag{161}$$

In conclusion, the fairness can be estimated by

$$\mathcal{FAA}_{avg}(P) \geq \mathcal{E}_E(\bar{\theta}) - \mathcal{E}_1(\bar{\theta}) \tag{162}$$

$$\geq \Big(\frac{E-1}{E} - \frac{L}{\mu E^2}\Big)R - \Big(1 + \frac{L(E-1)}{\mu E} - \frac{L^2}{\mu^2 E}\Big)B - \frac{2L}{\mu E}\sqrt{B(R - \frac{L}{\mu}B)} \tag{163}$$

$\square$

### C.3 PROOF OF DIVERGENCE REDUCTION

Here we prove the claim that the assumption $\mathcal{L}_E(\theta_e^*) - \mathcal{L}_E(\theta_E^*) \geq R$ is implied by a lower bound of the H-divergence (Zhao et al., 2022).

$$D_H(\mathcal{P}_e, \mathcal{P}_E) \geq \frac{LR}{4\mu} \tag{164}$$

*Proof.* Note that

$$D_H(\mathcal{P}_e, \mathcal{P}_E) = \frac{1}{2}\min_\theta\Big(\mathcal{L}_e(\theta) + \mathcal{L}_E(\theta)\Big) + \frac{1}{2}\Big(\mathcal{L}_e(\theta_e^*) + \mathcal{L}_E(\theta_E^*)\Big) \tag{165}$$

$$\leq \frac{1}{2}\Big(\mathcal{L}_e(\frac{\theta_e^* + \theta_E^*}{2}) + \mathcal{L}_E(\frac{\theta_e^* + \theta_E^*}{2})\Big) - \frac{1}{2}\Big(\mathcal{L}_e(\theta_e^*) + \mathcal{L}_E(\theta_E^*)\Big) \tag{166}$$

$$\leq \frac{1}{2} \times (\frac{1}{2}L\|\frac{\theta_E^* - \theta_e^*}{2}\|_2^2 \times 2) \tag{167}$$

$$= \frac{1}{8}L\|\theta_E^* - \theta_e^*\|_2^2 \tag{168}$$

Therefore,

$$\mathcal{L}_E(\theta_e^*) - \mathcal{L}_E(\theta_E^*) \geq \frac{\mu\|\theta_E^* - \theta_e^*\|_2^2}{2} \tag{169}$$

$$\geq \frac{\mu}{2}\frac{8D_H(\mathcal{P}_e, \mathcal{P}_E)}{L} = R. \tag{170}$$

$\square$

## D EXPERIMENTAL DETAILS

### D.1 EXPERIMENTAL SETUPS

Here we elaborate more details of our experiments.

**Machines.** We simulate the federated learning setup on a Linux machine with AMD Ryzen Threadripper 3990X 64-Core CPUs and 4 NVIDIA GeForce RTX 3090 GPUs.

**Hyperparameters.** For each FL experiment, we implement both FOCUS algorithm and FedAvg algorithm using SGD optimizer with the same hyperparameter setting. Detailed hyperparameter specifications are listed in Table 10 for different datasets, including learning rate, the number of local training steps, batch size, the number of training epochs, etc.

### D.2 ADDITIONAL EXPERIMENTAL RESULTS

**Histogram of loss on CIFAR.** Fig. 3 shows the surrogate excess risk of every agent trained with FedAvg and FOCUS on CIFAR dataset. For the outlier cluster that rotates 180 degrees (i.e., 2rd cluster), FedAvg has the highest test loss for the 9th agent, resulting in a high excess risk of 2.74.

Table 10: Dataset description and hyperparameters.

| Dataset | # training samples | # test samples | E | M | batch size | learning rate | local training epochs | epochs |
|---------|-------------------|----------------|---|---|------------|---------------|----------------------|--------|
| MNIST | 60000 | 10000 | 10 | 3 | 6000 | 0.1 | 10 | 300 |
| CIFAR | 50000 | 10000 | 10 | 2 | 100 | 0.1 | 10 | 200 |
| Yelp/IMDB | 56000/25000 | 38000/25000 | 10 | 2 | 512 | 5e-5 | 2 | 3 |

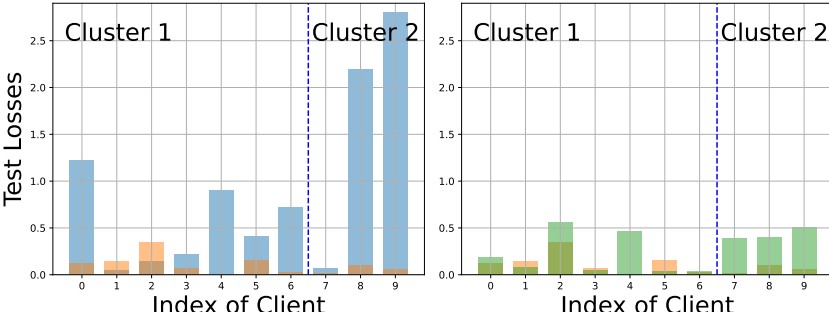

Figure 3: The excess risk of different agents trained with FedAvg (left) and FOCUS (right) on CIFAR dataset.

In addition, the agents in 1st cluster trained by FedAvg are influenced by the FedAvg global model and have high excess risk. On the other hand, FOCUS successfully identifies the outlier distribution in 2nd cluster, leading to a much lower excess risk among agents with a more uniform excess risk distribution. Notably, FOCUS reduces the surrogate excess risk for the 9th agent to 0.44.

**Comparison with existing fair FL methods.** We present the full results of existing fair federated learning algorithms on our data settings in terms of FAA. The results in Tables 11 and 12 show that FOCUS achieves the lowest FAA score compared to existing fair FL methods. We note that fair FL methods (i.e., q-FFL (Li et al., 2020b) and AFL (Mohri et al., 2019)) have lower FAA scores than FedAvg, but their average test accuracy is worse. This is mainly because they mainly aim to improve bad agents (i.e., with high training loss), thus sacrificing the accuracy of agents with high-quality data.

Table 11: Comparison of FOCUS and the existing fair federated learning algorithms on the rotated MNIST dataset.

| | FOCUS | FedAvg | q-FFL | | | | | AFL |
|---|-------|--------|-------|---|---|---|---|-----|
| | | | $q=0.1$ | $q=1$ | $q=3$ | $q=5$ | $q=10$ | $\lambda=0.01$ |
| Avg test accuracy | **0.953** | 0.929 | 0.922 | 0.861 | 0.770 | 0.731 | 0.685 | 0.885 |
| Avg test loss | **0.152** | 0.246 | 0.269 | 0.489 | 0.777 | 0.900 | 1.084 | 0.429 |
| FAA | **0.094** | 0.363 | 0.388 | 0.612 | 0.547 | 0.419 | 0.253 | 0.220 |

**Comparison with state-of-the-art FL methods.** We compare FOCUS with other SOTA FL methods, including FedMA (Wang et al., 2020), Bayesian nonparametric FL (Yurochkin et al., 2019) and FedProx (Li et al., 2020a). Specifically, the matching algorithm in (Yurochkin et al., 2019) is designed for only fully-connected layers, and the matching algorithm in (Wang et al., 2020) is designed for fully-connected and convolutional layers, while our experiments on CIFAR use ResNet-18 where the batch norm layers and residual modules are not considered in (Wang et al., 2020; Yurochkin et al., 2019). Therefore, we evaluate (Li et al., 2020a; Wang et al., 2020; Yurochkin et al., 2019) on MNIST with a fully-connected network, and (Li et al., 2020a) on CIFAR with a ResNet-18 model.

The results on MNIST and CIFAR in Tables 13 and 14 show that FOCUS achieves the highest average test accuracy and lowest FAA score than SOTA FL methods.

Table 12: Comparison of FOCUS and the existing fair federated learning algorithms on the rotated CIFAR dataset.

| | FOCUS | FedAvg | q-FFL | | | | | AFL |
| | | | $q = 0.1$ | $q = 1$ | $q = 3$ | $q = 5$ | $q = 10$ | $\lambda = 0.01$ |
|---|---|---|---|---|---|---|---|---|
| Avg test accuracy | **0.929** | 0.908 | 0.897 | 0.833 | 0.778 | 0.699 | 0.565 | 0.866 |
| Avg test loss | **0.217** | 0.262 | 0.306 | 0.704 | 0.876 | 1.139 | 1.263 | 0.466 |
| FAA | **0.365** | 0.537 | 0.661 | 0.542 | 0.525 | 0.494 | 0.421 | 0.619 |

Table 13: Comparison of FOCUS and other SOTA federated learning algorithms on the rotated MNIST dataset.

| | FOCUS | FedAvg | FedProx | | | FedMA | Bayesian Nonparametric |
| | | | $\mu = 1$ | $\mu = 0.1$ | $\mu = 0.01$ | | |
|---|---|---|---|---|---|---|---|
| Avg test accuracy | **0.953** | 0.929 | 0.908 | 0.927 | 0.929 | 0.753 | 0.517 |
| Avg test loss | **0.152** | 0.246 | 0.315 | 0.252 | 0.246 | 0.856 | 2.293 |
| FAA | **0.094** | 0.363 | 0.526 | 0.378 | 0.365 | 1.810 | 0.123 |

## E  BROADER IMPACT

This paper presents a novel definition of fairness via agent-level awareness for federated learning, which considers the heterogeneity of local data distributions among agents. We develop FAA as a fairness metric for Federated learning and design FOCUS algorithm to improve the corresponding fairness. We believe that FAA can benefit the ML community as a standard measurement of fairness for FL based on our theoretical analyses and empirical results.

A possible negative societal impact may come from the misunderstanding of our work. For example, low FAA does not necessarily mean low loss or high accuracy. Additional utility evaluation metrics are required to evaluate the overall performance of different federated learning algorithms. We have tried our best to define our goal and metrics clearly in Section 3; and state all assumptions for our theorems accurately in Section 4 to avoid potential misuse of our framework.

Table 14: Comparison of FOCUS and other SOTA federated learning algorithms on the rotated CIFAR dataset.

| | FOCUS | FedAvg | FedProx | | |
| | | | $\mu = 1$ | $\mu = 0.1$ | $\mu = 0.01$ |
|---|---|---|---|---|---|
| Avg test accuracy | **0.929** | 0.908 | 0.910 | 0.922 | 0.925 |
| Avg test loss | **0.217** | 0.262 | 0.282 | 0.245 | 0.239 |
| FAA | **0.365** | 0.537 | 0.698 | 0.700 | 0.654 |

