# OpenReview forum: "FOCUS: Fairness via Agent-Awareness for Federated Learning on Heterogeneous Data"
_ICLR.cc/2023/Conference — Submitted to ICLR 2023_

### Official Review · Reviewer_MUHj · 2022-10-20

**Confidence:** 3
**Correctness:** 2
**Technical Novelty And Significance:** 2
**Empirical Novelty And Significance:** 2
**Recommendation:** 3

**Clarity, Quality, Novelty And Reproducibility:**

### Clarity and Quality

The paper is generally clearly and well-written.

### Novelty

There is some novelty in the new definition of the fairness.

### Reproducibility

The reproducibility is good.

**Strength And Weaknesses:**

### Strengths

- The exposition is generally clear in terms of setting, algorithm, theoretical results.

- Theoretical guarantees (both convergence and fairness analysis) are provided.

- Several real datasets are adopted in experiments.


### Weaknesses
- The motivation for fairness in federated learning can be strengthened. To elaborate, the abstract does not seem to motivate why fairness is important or necessary in FL. Similarly, the introduction does not seem to clearly motivate why fairness is important in FL.

- The specific design choice for fairness via agent-awareness (i.e., the equity among the difference between their excess losses) can be motivated better. To elaborate, why is equity in any form a good notion of fairness? What are the motivating use cases for this notion of fairness?

- Related to the above point, in introduction it says
    > "we aim to define and enhance fairness by explicitly considering different contributions of heterogeneous agents"

    But it seems the heterogeneity is indirectly defined through the distributions of datasets instead of _explicitly_ via some concrete quantity to measure the heterogeneity. Moreover, it does not seem to provide a clear/explicit definition for the contributions. Following this, how does this fairness relate different levels of contributions of the clients?

- The usage of Bayes optimal error can be elaborated better. For instance, how does Bayes optimal error affect the analysis, especially for regression-based analysis? What would happen if Bayes optimal error is not used? A subsequent question, how are high-quality and low-quality data defined?

- The formalization of the heterogeneity of data can be justified better. In particular, the features of the datasets are assumed to be generated from Gaussian distributions with different means but identical and isotropic covariances. And each dataset has a true underlying mean vector. Is there justification/motivation for this setting (e.g., real use-cases where data are generated in this way, or approximately)? Then, is clustering performed w.r.t. the full model parameters (treated as a high-dimensional vector)? If so, how will the clustering affect the FL performance in such high-dimensional space?

- It seems some other related works (listed below) might also be relevant and can be empirically compared.

#### References

Tian Li et al, 2021, ICML, Ditto: Fair and Robust Federated Learning Through Personalization.

Xinyi Xu, Lingjuan Lyu, Xingjun Ma, Chenglin Miao, Chuan Sheng Foo, and Bryan Kian Hsiang Low. Gradient driven rewards to guarantee fairness in collaborative machine learning. Advances in Neural Information Processing Systems, 34:16104–16117, 2021.

I believe the second reference is included in the paper

**Summary Of The Paper:**

The paper studies fairness in federated learning where clients have heterogeneous local data. In particular, the paper introduces a new fairness definition/measure to leverage the excess risk at each local client. Subsequently, the paper proposes an algorithm that provably outperforms FedAvg in terms of this new fairness definition under certain conditions. Empirical results on several real datasets against some baselines are provided.

**Summary Of The Review:**

The motivation for the problem (i.e., fairness in FL) studied by this paper can be strengthened. There seems to be some mismatch between what is mentioned in the introduction and the main paper (e.g., contributions of agents are not explicitly/clearly defined). The approach (via Bayes optimal error) and setting (data heterogeneity) can be justified better. Overall, there is certainly merit to this paper, but in its current form, it is not ready to be published.

---

> ### Author Response · Authors · 2022-11-17
> **Response to Reviewer MUHj [1/3]**
>
> We thank the reviewer for the valuable comments and we provide our responses below.
>
> > Q1: The motivation for fairness in federated learning can be strengthened. To elaborate, the abstract does not seem to motivate why fairness is important or necessary in FL. Similarly, the introduction does not seem to clearly motivate why fairness is important in FL.
>
> Thanks for the helpful comment. Sorry about the confusion. In the abstract and introduction, we mainly discussed the limitation of existing fair FL approaches, which motivates our work to define more advanced fairness definition and optimization in FL. We have added more examples to motivate fairness in FL directly in our revision as well, following the reviewer’s suggestions. Specifically, existing work usually considers accuracy equity as fairness for different agents in FL, which is limited, especially under the heterogeneous setting, since it is intuitively “unfair” to enforce agents with high-quality data to achieve similar accuracy to those who contribute low-quality data.
>
> For example, considering the FL applications for medical analysis, some hospitals have high-resolution medical data and fine-grained labels, which cost a large amount of money to collect the data from advanced equipment and to crowdsource data labeling. In contrast, some hospitals may have low-resolution medical data and noisy labels.  In such a setting, high-quality agents may not be willing to participate in collaborative learning with low-quality agents because they could have achieved higher accuracy by standalone local training. **Therefore, a proper fairness notion is important to encourage agents to participate in FL and ensure fairness.** We added the above discussion in the introduction and abstract in our revision.
>
> > Q2: The specific design choice for fairness via agent-awareness (i.e., the equity among the difference between their excess losses) can be motivated better. To elaborate, why is equity in any form a good notion of fairness? What are the motivating use cases for this notion of fairness?
>
> Thanks for your suggestion. In general, fairness is usually defined as the protection of a specific attribute. Equity means each individual that joins the collaborative learning would not suffer from bad performance due to its identity. Previous work on fair FL considered equity of accuracy (q-FFL[1]), or proposed to optimize the performance of the worst agent (AFL[2]). However, as we stated in our paper (the 3rd paragraph in Section 1), these metrics are limited in heterogeneous data settings, as it is unfair for agents with high data quality to have the same accuracy as agents with low data quality, e.g., free riders, which compromise the performance of agents with high data quality.
>
> This motivates us to define the FAA metric that enforces **equity of excess risks** among agents, following the philosophy that each agent should **"gain the same"** from participating in federated learning.  For instance, when a local agent has low-quality data, although the corresponding utility loss would be high, the Bayes error of such low-quality data is also high, and thus the excess risk of the user is still low, enabling the agents with high-quality data to achieve low utility loss for fairness. Under FAA, the performance of agents with high-quality data will not be sacrificed just due to the existence of agents with low-quality data. The motivating use cases for the proposed fairness include FL under heterogeneous settings, especially with different data qualities. We have added the corresponding discussion in our revision, and thank you for pointing it out.
>
> Reference:
> - [1] Li, Tian, et al. "Fair resource allocation in federated learning." arXiv preprint arXiv:1905.10497 (2019).
> - [2] Mohri, Mehryar, Gary Sivek, and Ananda Theertha Suresh. "Agnostic federated learning." International Conference on Machine Learning. PMLR, 2019.

---

> ### Author Response · Authors · 2022-11-17
> **Response to Reviewer MUHj [2/3]**
>
>
> > Q3: it seems the heterogeneity is indirectly defined through the distributions of datasets instead of explicitly via some concrete quantity to measure the heterogeneity. Moreover, it does not seem to provide a clear/explicit definition for the contributions. Following this, how does this fairness relate different levels of contributions of the clients?
>
> Thanks for the comment. Indeed here we define the contribution of heterogeneous agents by their data quality, which can be represented by the Bayes optimal loss of the data by definition. Concretely,  the **data quality** of each agent is quantitively measured by its **Bayes optimal loss**, as shown in Eq. 3. Agents that collect data with large noises or erroneous labels typically have high Bayes optimal error. We have added corresponding discussions in Section 3.1.
>
>
> > Q4: The usage of Bayes optimal error can be elaborated better. For instance, how does Bayes optimal error affect the analysis, especially for regression-based analysis? What would happen if Bayes optimal error is not used? A subsequent question, how are high-quality and low-quality data defined?
>
> Thanks for the comment. Here the high-quality data is defined as the data distribution that results in a low Bayes optimal error. Note that there are other ways to define or measure the data quality or other properties as well, which can also be plugged into our fairness definition.
>
> The reasons for using Bayes optimal error in our fairness metric can be explained in the following two aspects.
> - (1) The motivation for using Bayes optimal error. As explained in Section 3.1, we measure the contribution of each agent by the Bayes optimal error with respect to its data distribution. If Bayes optimal error is not used, the fairness metric reduces to the “loss parity” (similar metric as “accuracy parity”), where agents that collect high-quality data (i.e., with low Bayes optimal error) will be compromised by agents with low-quality data or “free riders”, which is not fair for agents with high-quality data.
> - (2) For the fairness analysis in theorem 3 and 4. The usage of Bayes optimal error enables an upper and lower bound for our fairness analysis. By subtracting the Bayes optimal error from the loss function, we can describe the excess risk of each agent using the distance from the current weight of the model to the optimal solution (accurately for linear models in theorem 3, lower and upper bounds for strongly convex loss functions in theorem 4). This step is important for our analysis. If Bayes optimal error is not used, we need to adjust the calculation of the loss differences among agents as we did in Appendix C accordingly.
>
>
> > Q5: Better justification of data settings, especially "Gaussian distributions with different means but identical and isotropic covariances"  Is clustering performed w.r.t. the full model parameters (treated as a high-dimensional vector)? If so, how will the clustering affect the FL performance in such high-dimensional space?
>
> Thank you for the helpful comment. We first clarify that in our paper we do not only discuss Gaussian distributed datasets in our analysis. In fact, we consider 1) linear models with Gaussian distributions, and 2) models with strongly convex loss functions when the data distributions are not necessarily Gaussian. We aim to analyze the simple and generalized settings step by step, so that it is easier to get the intuition behind our analysis. We include the analysis for case 1 because its conclusion is concise and illustrative. Moreover, it can be viewed as a distributed version of mixed linear regression, which is a widely studied optimization problem [Yi et al., ICML 2014].
>
> Yes, the reviewer is correct that we treat the full model parameters as a high-dimensional vector. The dimensionality of model parameters does not affect our analysis, as the bounds derived in our theorems are independent of the model dimensionality. As shown in the experimental results, with the large models (ResNet-18 for CIFAR, BERT-based model for text data), FOCUS still outperforms FedAvg and existing fair FL methods.
>
> Reference:
>
> [Yi et al., ICML 2014] Yi, Xinyang, Constantine Caramanis, and Sujay Sanghavi. "Alternating minimization for mixed linear regression." International Conference on Machine Learning. PMLR, 2014.

---

> ### Author Response · Authors · 2022-11-17
> **Response to Reviewer MUHj [3/3]**
>
>
> > Q6: It seems some other related works might also be relevant and can be empirically compared.
>
> Thanks for your suggestions. We add the comparison to Ditto [1]  and CGSV [2] as suggested by the reviewer in Table 1 on four datasets and Table 3. We also added the discussion for Ditto and CGSV in Section 2 Related Work.
>
> The experimental results in Table 1 on four datasets and Table 3 on MNIST show that FOCUS still achieves the strongest fairness among agents in terms of FAA compared to all baselines.
> - Ditto proposes to train personalized models with global model regularization to improve the fairness for personalized models, while FOCUS leverages the soft cluster assignment, which is personalized for each client as well, with ensemble prediction from clustering models. The experimental results show the superiority of the EM-based clustering mechanism for FAA fairness under heterogeneous data.
> - CGSV approximates the agent contribution by gradient cosine similarity and uses personalized gradient sparsification level for each agent to improve fairness. However, CGSV hurts the accuracy by sparsification, and the gradient cosine similarity did not consider the local data distribution for agent contribution, thus still incurring high FAA (i.e., low fairness).
>
>
>
> Reference:
> - [1] Tian Li et al, 2021, ICML, Ditto: Fair and Robust Federated Learning Through Personalization.
> - [2] Xinyi Xu, Lingjuan Lyu, Xingjun Ma, Chenglin Miao, Chuan Sheng Foo, and Bryan Kian Hsiang Low. Gradient driven rewards to guarantee fairness in collaborative machine learning. Advances in Neural Information Processing Systems, 34:16104–16117, 2021.

---

> > ### Comment · Reviewer_MUHj · 2022-11-19
> > **Post rebuttal**
> >
> > I thank the authors for the detailed response. While some of my questions are addressed, a few remain.
> >
> > - Regarding the motivation of fairness and the specific formulation of fairness:
> >
> >     > In general, fairness is defined as the protection of a specific attribute
> >
> >     The relevance of this notion of fairness to this paper is not so clear. Exactly which "specific attribute" does FAA aim to protect?
> >
> >     > However, the fairness analysis in FL under heterogeneous data distributions is still lacking.
> >
> >     The paper by Li et al in ICML2021 (Ditto) and Xu et al in NeurIPS2021 (CGSV) seem to both consider fairness under heterogeneous data distributions. Precisely in which way is this formulation of fairness (i.e., FAA) superior than these existing methods?
> >
> >     > following the philosophy that each agent should “gain the same” from participating in FL
> >
> >     To me, the so-called ''gain the same'' philosophy seems contradictory to the used example where agents may have data of different qualities. Specifically, if these agents have data of different qualities, why is it fair to allow them to gain the same? Shouldn't the agents with data of higher qualities gain more, in order to be fair?
> >
> > - Regarding heterogeneity. I appreciate the clarification that the contribution is via the Bayes optimal loss and that the Bayes optimal losses are different across agents due to the heterogeneity in agents' data:
> >
> >     I wonder if it indeed is able to explicitly/precisely capture the heterogeneity (in data distributions) without explicitly defining heterogeneity through teh data distributions. For instance, is it possible for two different data distributions to have the exact same Bayes optimal loss? If so, is this a case of heterogeneity?
> >
> >     This is to point out that, despite the appealing properties of Bayes optimal loss, using it to frame the approach towards heterogeneity might be lacking in some aspect and seems to suggest some additional careful thinking can be helpful.
> >
> >
> > - Regarding the experiments on how to approximate the Bayes optimal loss.
> >
> >     > Specifically, we train a centralized model based on the subset of agents with similar data distributions (i.e., the same ground-truth cluster) and use it as a surrogate to approximate the Bayes optimum.
> >
> >     While I understand that for the purpose of experiments, a centralized model can be trained and used as a surrogate, it seems to make it difficult for this way of evaluating the fairness of the algorithm to be actually used in practice. Is there an evaluation of fairness/FAA that is more practical, otherwise, it might take away the practical usefulness of this fairness notion (i.e., FAA).

---

> ### Author Response · Authors · 2022-11-21
> **Response to Post rebuttal for Reviewer MUHj [1/3]**
>
> We thank the reviewer for the insightful feedback, and we provide our detailed answers below and will update our final version accordingly.
>
> > Q1 The relevance of this notion of fairness to this paper is not so clear. Exactly which "specific attribute" does FAA aim to protect?
>
> Sorry for the confusion. We meant that in the general ML setting (centralized setting), fairness was proposed to protect specificity attributes (i.e., gender, age). However, in the FL setting, the specific fairness here aims to protect “agent-level” fairness rather than the “attribute-level” (the attribute-level fairness can still be analyzed in FL for the final global model, which shares similar analysis with the centralized setting, but it is not the focus of existing FL fairness literature or our work). We will make this sentence clearer in our revision.
>
> > Q2 following the philosophy that each agent should “gain the same” from participating in FL. To me, the so-called ''gain the same'' philosophy seems contradictory to the used example where agents may have data of different qualities. Specifically, if these agents have data of different qualities, why is it fair to allow them to gain the same? Shouldn't the agents with data of higher qualities gain more, in order to be fair?
>
> Thanks for the comment. The reviewer has the correct intuition. Actually, based on the FAA fairness definition for the equality of excess risk, the lower the Bayes optimal error (i.e., higher quality data), **the lower the final model loss the agent can achieve**.
> Therefore, the higher-quality agents are allowed to achieve better model performance (i.e., lower loss) under FAA. This is to say, from the **absolute performance** perspective, the higher-quality agents can achieve higher utility (lower loss) to be fair,  which is exactly what the reviewer suggests.
> From the **relative performance** (i.e., excess risk) perspective, all agents should gain the same under FAA for fairness since this relative performance is on purposely defined to eliminate the impact of their data quality by subtracting the Bayes error, and thus the relative performance should be similar to be fair.

---

> ### Author Response · Authors · 2022-11-21
> **Response to Post rebuttal for Reviewer MUHj [2/3]**
>
> > Q3 The paper by Li et al in ICML2021 (Ditto) and Xu et al in NeurIPS2021 (CGSV) seem to both consider fairness under heterogeneous data distributions. Precisely in which way is this formulation of fairness (i.e., FAA) superior than these existing methods?
>
> Thanks for the comment. We illustrate in detail below why FAA provides a more general and precise fairness definition in FL under the heterogeneous setting. We will add the discussions in our related work as well.
>
> (1) Discussion on Ditto
> - Ditto considers fairness based on accuracy parity (i.e., the more uniform the accuracy among agents is, the fairer it is for agents) and provides fairness analysis in the form that the optimal solution of Ditto minimizes the variance of test loss across agents. However, as we mentioned in Section 3.1, **the notion of accuracy parity does not consider the important properties of heterogeneous data (especially different data qualities)** because the performance of agents with high-quality data could be sacrificed by the agents with low-quality data, which motives us to propose FAA in the form of excess risk quality, where agents with higher data quality (i.e., lower Bayes optimal error) are allowed to achieve lower losses for fairness. Moreover, we provide formal fairness analysis for FOCUS and FedAvg based on FAA (Theorem 3 and Theorem 4).
>
> (2) Discussion on CGSV
> - CGSV studies the notion of fairness that higher-quality parameter updates should be finally rewarded with lower training loss, but it does not consider the heterogeneous data distributions in its fairness analysis. Specifically, CGSV analyzes that the higher the agent’s importance coefﬁcient (estimated by gradient similarity), the lower its training loss, which is rather direct because CGSV explicitly uses the important coefficient to sparsify the gradient as a reward to the agent, and the sparsifying directly impacts the model performance.  However, such **importance coefﬁcient based on gradients agreement does not necessarily reflect the importance/quality of data distribution**. For example, the existence of a large number of low-quality agents can force them to have a larger gradient similarity, thus a higher important coefficient. Moreover, CGSV uses sparsifying among agents to enforce fairness, which hurts the model accuracy, as we show in Table 1.
> - In contrast, we define fairness in the form of excess risk quality, where agents with higher data quality (i.e., lower Bayes optimal error) are allowed to achieve lower losses. Moreover, we use clustering to achieve fairness by reducing intra-cluster heterogeneity, which does not require additional tricks like sparsification. In our fairness analysis, we study the **relationship between the agent’s excess risk and its data distribution**. This is more challenging than the gradient analysis in CGSV because our analyses are built on assumptions of data distribution only (e.g., total variance distance for Theorem 4), while CGSV makes assumptions of an “importance coefficient” related to historical gradient information.
>
> > Q4 I wonder if it indeed is able to explicitly/precisely capture the heterogeneity (in data distributions) without explicitly defining heterogeneity through teh data distributions. For instance, is it possible for two different data distributions to have the exact same Bayes optimal loss? If so, is this a case of heterogeneity?
> This is to point out that, despite the appealing properties of Bayes optimal loss, using it to frame the approach towards heterogeneity might be lacking in some aspect and seems to suggest some additional careful thinking can be helpful.
>
> Thanks for the comment, and sorry for the confusion. Indeed, Bayes optimal loss does not fully describe the **heterogeneity of distribution**, i.e., two agents with different data distributions may have the same Bayes optimal loss. However, to define fairness among agents, we only need to capture the **heterogeneity of data quality/contribution**. The Bayes optimal loss accurately defines the heterogeneous contribution among agents, i.e., two agents with heterogeneous data quality/contribution mean they have different Bayes optimal loss. We will clarify this in our revision.

---

> ### Author Response · Authors · 2022-11-21
> **Response to Post rebuttal for Reviewer MUHj [3/3]**
>
> > Q5 Regarding the experiments on how to approximate the Bayes optimal loss. While I understand that for the purpose of experiments, a centralized model can be trained and used as a surrogate, it seems to make it difficult for this way of evaluating the fairness of the algorithm to be actually used in practice. Is there an evaluation of fairness/FAA that is more practical, otherwise, it might take away the practical usefulness of this fairness notion (i.e., FAA).
>
> Thanks for the valuable comment, and sorry for the confusion. In our paper, we explicitly approximate the Bayes optimal error with the performance error of the centralized models for evaluation, and we show that FOCUS achieves much higher fairness so that practitioners could directly use the FOCUS algorithm to train fair FL models, regardless of the evaluation approaches. We note that such a setting for evaluation is practical and can be directly used in different ways. For instance, in practice, some local agents usually have a large amount of local data, and thus we can evaluate the approximated Bayes optimal error based on the agent’s local model trained and evaluated on its own data for the fairness evaluation purpose. Alternatively, we can also use the average (or other aggregation) of the local model’s loss within a cluster to approximate the Bayes optimal loss for such a distribution, which usually should be very close to the loss of one local model since these agents contain data from the similar distribution.
>
> We acknowledge that the approximation of Bayes optimal loss for a given data distribution could be challenging in practice and still an open problem, but since in our fairness evaluation, we only need to calculate the “relative difference” rather than the absolute values for fairness comparison, our approximation based evaluation is enough to serve our comparison purpose and can be done in a practical way.
>
>  We will add such discussions in our revision to make it more clear.

---

> ### Author Response · Authors · 2022-11-29
> **Following Up with Reviewer MUHj**
>
> Thanks again for your thoughtful review. We really appreciate your insightful comments that helped us to eliminate the ambiguity of some of our statements, and we will improve the statements in our revision to make them clear.  Do our responses help address your comments?
>
> Following the valuable suggestions from all reviewers, we also summarized our updates in the comment “Paper Revision Summary”, and we are happy to answer other questions and provide related evaluations if there are additional concerns. Please do not hesitate to contact us if you have questions, and we would appreciate the opportunity to engage further. Thanks!

---

### Official Review · Reviewer_wM5N · 2022-10-24

**Confidence:** 3
**Correctness:** 4
**Technical Novelty And Significance:** 2
**Empirical Novelty And Significance:** 2
**Recommendation:** 6

**Clarity, Quality, Novelty And Reproducibility:**

Overall the paper is written nicely and the structure was clear to follow. There was some concern on the comparisons and evaluations w.r.t. other baselines in the prior works. The implementation details and code are provided.

**Strength And Weaknesses:**

Strength:
- Learning fair models in federated setting with heterogeneous agents is an important research problem. The proposed fairness criteria minimizes the maximal difference of excess risks for any pair of agents which is a natural criteria to use.
- The FOCUS is based on the Expectation-Maximization algorithm which is very intuitive.
- Both the theoretical convergence guarantees and experimental comparisons are studied carefully, and the experimental results demonstrate the effectiveness of the FOCUS algorithm in minimizing the proposed fairness loss.

Weakness:
- First concern is about the scalability of the proposed fairness criteria and algorithm. The proposed metric minimizes the maximal difference of excess risks for any pair of agents, whose evaluation requires looking at all the pairs of agents --which grows exponentially with the number of agents. It seems that even evaluating such a fairness criteria is computationally-expensive with a large number of agents.
- Following the scalability concern, the EM algorithm also tends to have a slow convergence.
- The experimental comparisons are comparing the proposed algorithm ---which is designed to minimize the proposed fairness loss ---to other methods which are not designed for that. I think a fair set of comparisons should include the performance of FedAvg if that is also using the clusters that FOCUS ends up using; and evaluating the results of FOCUS on the fairness criterion proposed in other works, e.g. the agnostic loss in AFL (Mohri et al., 2019). Moreover the runtime and computational costs should be compared as well. From the current evaluations, it is hard to conclude that FOCUS out-performs the existing methods. E.g. FedAvg is not using any clustering and outputs one single model; but FOCUS is allowed to return M models to the agents.

Typos:
- Equation (2), $\Epsilon$ missing the subscript $e_1$

**Summary Of The Paper:**

This paper studies the problem of training fair models in federated learning with heterogeneous agents. The authors proposed a new fairness metric which is the maximal difference of excess risks for any pair of agents. The authors then proposed an algorithm which is based on the Expectation-Maximization algorithm, that clusters the similar agents into groups and minimize the fairness-violation loss.

Theoretically the convergence guarantee for the proposed algorithm was provided for linear agents, and non-linear agents with smooth and strongly convex loss functions. Empirically the authors compared the proposed algorithm with FedAvg, q-FFL (Li et al., 2020b) and AFL (Mohri et al., 2019), showing that the proposed algorithm performs better under the proposed fairness criteria.

**Summary Of The Review:**

Overall this paper studies an important problem ---FL with heterogeneous agents. The proposed fairness metric and algorithm are quite intuitive. The experimental comparisons are missing some necessary evaluations. I also have some concern on the scalability of the proposed approach.


---------
Post-rebuttal: I thank the authors for the detailed response to my questions and concerns! The additional clarifications and comparisons addressed my concerns on the scalability and computation cost of the proposed methods. Therefore I'd like to improve my score based on the author feedback.

---

> ### Author Response · Authors · 2022-11-17
> **Response to Reviewer wM5N [1/3]**
>
> We thank the reviewer for the valuable comments and we provide our responses below.
>
> > Q1: scalability of the proposed fairness criteria and algorithm. The proposed metric minimizes the maximal difference of excess risks for any pair of agents, which grows exponentially with the number of agents. It seems that even evaluating such a fairness criteria is computationally-expensive with a large number of agents.
>
> Thanks for the helpful comment.
>
> (1) The **scalability of the proposed fairness criteria FAA.**
>
> Regarding the scalability of our FAA metric, we note that the evaluation of FAA is scalable and does not take exponential time. To calculate the maximal difference of excess risks for any pair of agents, **it suffices to calculate the difference between the maximal and minimal** per-client excess risk, and we don’t need to calculate the difference for any pairs of agents. The overall time cost is **linear** to the number of agents.
>
> In addition, we compare the computation time (averaged over 100 trials) of FAA and existing fairness criteria (i.e., Accuracy Parity and Agnostic Loss) under 10 clients and 100 clients on MNIST. The below table (also as Table 4 in appendix A)  shows that the computation of FAA is efficient even with a large number of agents.  Moreover, calculating the difference between maximal and minimal excess risk (i.e., FAA) is even faster than calculating the standard deviation of the accuracy between agents  (i.e., Accuracy Parity).
>
>
> |                  | 10 clients       | 100 clients     |
> |------------------|------------------|-----------------|
> | Accuracy Parity  | 4.70 e-05 second | 6.48e-05 second |
> | Agnostic loss    | 9.41 e-07 second | 3.92e-06 second |
> | FAA              | 6.09 e-06 second | 4.27e-05 second |
>
> (2)  Next, we study the **scalability of the proposed fair algorithm FOCUS**.
>
> We evaluate the performance and fairness of FOCUS and existing methods under 100 clients on MNIST. The below table (also as Table 3 in appendix A)  shows that FOCUS achieves the best fairness measured by FAA and Agnostic Loss, higher test accuracy, and lower test loss than Fedavg and existing fair FL methods.  We also report the test accuracy and test loss of different methods over FL communication rounds on MNIST 100 clients in Figure 2 (b) in Appendix A. The results show that FOCUS converges faster and achieves higher accuracy and lower loss than other methods.
>
> |                | FOCUS  | FedAvg | q-FFL  | AFL    | Ditto  | CGSV   |
> |----------------|--------|--------|--------|--------|--------|--------|
> | Avg test acc   | **0.9533** | 0.9236 | 0.8371 | 0.8813 | 0.9351 | 0.8691 |
> | Avg test loss  | **0.1570** | 0.2571 | 0.5668 | 0.4355 | 0.2206 | 0.6294 |
> | FAA            | **0.5605** | 1.0652 | 1.5055 | 0.8901 | 0.7459 | 1.2935 |
> | Agnostic loss  | **0.5028** | 0.8894 | 1.4227 | 0.7767 | 0.6200 | 1.5133 |
>
> We have added the above discussion in Appendix A.
>
> > Q2: Following the scalability concern, the EM algorithm also tends to have a slow convergence.
>
> Thank you for the helpful comment, and we clarify that FOCUS algorithm converges fast. Theoretically, as stated in Theorem 1 and 2, FOCUS algorithm converges linearly under mild conditions. Moreover, the clustering of agents reduces the variance of model weights when doing aggregation, which makes FOCUS converge faster.
>
> We empirically study the convergence speed of our EM algorithm (FOCUS) and report the test accuracy and test loss of different methods over FL communication rounds on MNIST and CIFAR with 10/100 clients in  Figure 2 (a-c) in Appendix A. The results show that FOCUS converges faster and achieves higher accuracy and lower loss than other methods.

---

> ### Author Response · Authors · 2022-11-17
> **Response to Reviewer wM5N [2/3]**
>
> > Q3: I think a fair set of comparisons should include the performance of FedAvg if that is also using the clusters that FOCUS ends up using ….. From the current evaluations, it is hard to conclude that FOCUS out-performs the existing methods. E.g. FedAvg is not using any clustering and outputs one single model; but FOCUS is allowed to return M models to the agents.
>
> Thanks for the helpful comment. In our evaluation we use the standard FedAvg protocol (McMahan et al., 2017) for comparison. Following the reviewer’s suggestion, we construct a new evaluation by combining the clustering and Fedavg together (i.e., FedAvg-HardCluster), as another baseline.
> To compare the performance between FOCUS and FedAvg-HardCluster, we consider two scenarios:
> - **Scenario 1**: underly clusters are clearly separatable, where each cluster contains samples from one distribution, which is the setting used in our paper.
> - **Scenario 2**: underlying clusters are not separatable, where each cluster has 80%, 10%, and 10% samples from three different distributions, respectively. For example, the first underlying cluster contains 80% samples without rotation, 10% samples rotating 90 degrees, and 10% samples rotating 180 degrees.
>
> The below table (also table 8 in Appendix A) presents the results of FOCUS and FedAvg-HardCluster on Rotated MNIST under two scenarios.
> - Under scenario 1, the accuracy of FOCUS and FedAvg-HardCluster is similar, and FOCUS achieves better fairness in terms of FAA. The results show that the hard clustering for FedAvg-HardCluster is as good as the soft clustering for FOCUS when the underlying clusters are clearly separable, which verifies that clustering is one of the key steps in FOCUS, and it aligns with our hypothesis for fairness under heterogeneous data.
> - Under scenario 2, FOCUS achieves higher accuracy and better FAA fairness than FedAvg-HardCluster. The results show that when underly clusters are not separatable, soft clustering is better than hard clustering since each agent can benefit from multiple cluster models with the soft $\pi$ learned from the EM algorithm in FOCUS. We add the discussion in Appendix A in our revision.
>
> |                 | Scenario 1  |  | Scenario 2 |   |
> |-----------------|-------------|--------------------------------------------|------------|----------------------------------------|
> |                 | FOCUS       | FedAvg-HardCluster                         | FOCUS      | FedAvg-HardCluster                     |
> | Avg test acc    | 0.953       | **0.954**                                  | **0.814**  | 0.812                                  |
> | Avg test loss   | 0.152       | 0.152                                      | **1.168**  | 1.244                                  |
> | FAA             | **0.094**   | 0.099                                      | **0.449**  | 0.459                                  |
> | Agnostic loss   | 0.224       | 0.224                                      | **1.333**  | 1.397                                  |
>
> We have added the above discussion in Appendix A.
>
> > Q4: evaluating the results of FOCUS on the fairness criterion proposed in other works, e.g. the agnostic loss in AFL (Mohri et al., 2019).
>
> Thanks for the insightful suggestion. We report the agnostic loss in AFL (Mohri et al., 2019) in Table 1 and Table 3 in our revisoin. The results show that FOCUS achieves the lowest agnostic loss compared to all baselines.

---

> ### Author Response · Authors · 2022-11-17
> **Response to Reviewer wM5N [3/3]**
>
> > Q5:  runtime and computational costs should be compared as well.
>
>
> Thanks for the valuable suggestions.
>
> (1) In terms of **computation costs**,  we report the number of communication rounds that each method takes to achieve targeted accuracy on MNIST and CIFAR in the below tables (also as Table 5 and Table 6 in Appendix A). We note that FOCUS requires a significantly smaller number of communication rounds than FedAvg, q-FFL, and AFL on both datasets, which demonstrates the small costs required by FOCUS.
>
> The number of communication rounds to reach target performance on MNIST
>
> |        | 70%   | 80%    | 85%    | 90%    |
> |--------|-------|--------|--------|--------|
> | FOCUS  | **9** | **16** | **20** | **29** |
> | FedAvg | 10    | 51     | 88     | 177    |
> | q-FFL  | 28    | 151    | 261    | >300   |
> | AFL    | 16    | 94     | 180    | >300   |
>
>
>
> The number of communication rounds to reach target performance on CIFAR
> |        | 70%    | 80%    | 85%     | 90%     |
> |--------|--------|--------|---------|---------|
> | FOCUS  | **69** | **92** | **112** | **135** |
> | FedAvg | 81     | 103    | 129     | 164     |
> | q-FFL  | 102    | 159    | >250    | >250    |
> | AFL    | 116    | 159    | 241     | >250    |
>
>
> (2) In terms of **runtime**, we report the training time for one FL round (averaged over 20 trials) as well as inference time (averaged over 100 trials) in the below table (also as Table 7 in Appendix A).
> - Since the local updates and sever aggregation for different cluster models can be run in parallel, we observe that FOCUS has a similar training time compared to FedAvg, q-FFL, and AFL, which train one global FL model.
> - For the inference time, FOCUS is slightly slower than existing methods by about 0.17 seconds due to the ensemble prediction of all cluster models at each client. However, we note that such cost is negligible, and the forward passes of different cluster models for the ensemble prediction can also be made in parallel to further reduce the inference time.
> |        | Training time per FL round (second) | Inference time (second) |
> |--------|-------------------------------------|-------------------------|
> | FOCUS  | 6.59                                | 0.28                    |
> | FedAvg | 6.23                                | 0.12                    |
> | q-FFL  | 6.31                                | 0.11                    |
> | AFL    | 6.24                                | 0.12                    |
>
> We have added the above discussion in Appendix A.
> > Q6: Typos: Equation (2), \Epsilon missing the subscript e1
>
> Thanks for pointing it out! We have corrected it in our revision.

---

> ### Author Response · Authors · 2022-11-29
> **Following Up with Reviewer wM5N**
>
> We thank the reviewer again for your valuable review. Here we want to follow up to make sure your concerns are addressed before the discussion period ends.
>
> Following the valuable suggestions from all reviewers, we also summarized our updates in the comment “Paper Revision Summary”, and we are happy to answer other questions and provide related evaluations if there are additional concerns. Please do not hesitate to contact us if you have additional questions. Thanks!

---

### Official Review · Reviewer_p3Qb · 2022-10-27

**Confidence:** 3
**Correctness:** 4
**Technical Novelty And Significance:** 3
**Empirical Novelty And Significance:** 3
**Recommendation:** 6

**Clarity, Quality, Novelty And Reproducibility:**

The work and definition of fairness are original. Results are supported by theorem and proofs as well as experiments.

**Strength And Weaknesses:**

Strength:
- the motivation of the paper is clear, and important. It is unfair to give the same outcomes and benefits to agents that barely contribute to the model learned by the federated learning algorithm and simply free-ride. Further, the presence of agents with low-quality data can further compromise the performance of agents with high-quality data.
- the formal definition of fairness is an interesting proxy to address this issue, by looking at the excess risk of the model seen by population e compared to the best model population e could train on their own data. In turn, if this excess risk is low, it guarantees that a population with low-quality data sees a model $\theta_e$ that cannot be (much) better than the low-quality model they could have trained themselves.
- the techniques are interesting, with a 2-step approach: i) clustering agents with similar data distribution together (so that agents with accurate/high-quality data are for example clustered together), then ii) give different weights to different agents as a function of their cluster in the loss minimization problem
- The authors provide theoretical statement showing that their algorithm performs well for a reasonable/common class of losses (linear + smooth and strongly convex)
- The authors provide convincing experiments showing that their approach leads to better fairness, but also overall model performance, than previous work. The results are especially good when the data is synthetically generated according to the assumptions of the paper, but the experiments also show that their approach does better than FedAvg and previous work even on real data.

Weaknesses:
- I found the notations of the main theorem a bit confusing, with K being the number of rounds (which is what I think would typically be called T), and T the number of updates in each round.
- Theorem 4 and the fairness analysis is a bit hard to understand in the case of smooth + strongly convex losses. It is also a bit weak in that i only looks at 2 clusters. However, this weakness is partially made up for through the experiments.
- It is possible I missed this, but the authors seem to assume that M is known in advance, since it is provided as an input to the algorithm. How can one estimate/upper-bound the number of clusters? This seems that it would require to run the clustering first, before actually knowing M, and evaluating how close the data distributions are in each cluster.

**Summary Of The Paper:**

The paper considers a federated learning setting. The idea and novelty of the paper consist in taking fairness considerations into account. Namely, agents that contribute a lot of/little data or lower/higher quality data should not be treated the same way.

**Summary Of The Review:**

I think this is a good paper that studies an important problem and provides a comprehensive (both theoretically and practically) solution to said problem, and as such recommend acceptance into the program

---

> ### Author Response · Authors · 2022-11-17
> **Response to Reviewer p3Qb**
>
> We thank the reviewer for the valuable comments and we provide our responses below.
>
> > Q1. Some notations are confusing with K being the number of rounds (which is what I think would typically be called T), and T the number of updates in each round.
>
> Thanks for your advice. We have swapped the notations of K and T following the suggestion in the revision to use T to denote the number of communication rounds.
>
> > Q2. Theorem 4 and the fairness analysis is a bit hard to understand in the case of smooth + strongly convex losses.
>
> Thank you for the insightful comment. To better understand Theorem 4 and the fairness analysis, we provide a remark in Section 4.2 that explains its meaning with simplified expressions (equation 24), i.e., FOCUS is guaranteed to achieve better FAA fairness than FedAvg when the number of agents is sufficiently high.
>
> > Q3. The weakness of only considering 2 clusters in fairness analyses.
>
> Thanks for the comment. In the fairness analysis, we focus on the case with M=2 underlying clusters and a single outlying agent because we want to use this simple but representative case study to show how agent clustering helps improve fairness in FL. Note that in theorem 3 and 4, we prove that FOCUS achieves **strictly higher fairness consistently** than FedAvg when $M=2$. When $M \geq 2$, there could exist some corner cases, which are mainly due to the properties of the clustering algorithm itself rather than the fairness analysis. We have added the discussions in our revision.
>
> > Q4. The weakness that the number of underlying clusters M is predefined.
>
> Thanks for the helpful comment. The performance of FOCUS would not be harmed if the selected number of clusters is larger than the number of underlying clusters since the superfluous clusters would be useless (the corresponding soft cluster assignment $\pi$ goes to zero). On the other hand, when the selected number of clusters is smaller than the number of underlying clusters, FOCUS would converge to a solution when some clusters contain agents from more than one underlying cluster.
>
> Empirically, in the below table (also as Table 9 in Appendix A), we have 3 true underlying clusters while we set M=1,2,3,4 in our experiments, and we see that when M=3,4, FOCUS achieves similar accuracy and fairness, which verifies our hypothesis that the superfluous clusters would become useless. When M=2, FOCUS even achieves the highest fairness, which might be because one cluster benefits from the shared knowledge of multiple underlying clusters. When M=1, FOCUS reduces to FedAvg, which does not have the clustering mechanism, leading to the lowest accuracy and fairness under heterogeneous data.  We have added the discussion in Appendix A.
>
>
> |                               | M=1 (FedAvg) | M=2        | M=3          | M=4        |
> |-------------------------------|--------------|------------|--------------|------------|
> | Avg test acc                  | 0.929        | 0.952     |  **0.953**    | **0.953** |
> | Avg test loss                 | 0.246        | 0.167     | **0.152** | 0.153     |
> | FAA                           | 0.363        | **0.079** | 0.094      | 0.091     |
> | Agnostic loss | 0.616     | 0.272     | 0.224      | **0.223** |

---

> ### Author Response · Authors · 2022-11-29
> **Following Up with Reviewer p3Qb**
>
> Thank you again for your thoughtful review. Do our responses help address your comments?
> As the end of the discussion is approaching, we would appreciate the opportunity to engage further if needed.
>
> Following the valuable suggestions from all reviewers, we summarize our updates in the comment “Paper Revision Summary”, and we are happy to further improve the paper if there are additional comments. Thanks!

---

> ### Comment · Reviewer_p3Qb · 2022-11-29
> **I am very sorry for the delayed answer and would like to thank the authors for the reminder**
>
> Thank you very much! I think the response was helpful. To go through it point by point:
> 1) Thank you for the notational change!
> 2) I find the clarification to be helpful and to shed more insights about the guarantees of FOCUS in this case.
> 3 and 4) I think it is useful to show in the experiments what happens when M is taken to be both larger than 2 and larger than the number of underlying clusters. It seems to show some evidence that having an upper bound on M is enough and that the framework of the paper can handle more than 2 clusters. However, M = 3 with an upper bound of 4 remains relatively small, so I think this could potentially be pushed a bit further to provide further evidence here.
>
> At the moment I still think this is an interesting paper, but I think I will still keep my score at 6 given that I think the major issue is with M being small and unknown and more evidence could be provided there.

---

### Author Response · Authors · 2022-11-17
**Paper Revision Summary**

We thank all the reviewers for their insightful questions and suggestions! Below is a summary of the major updates in our revision:
- [Abstract, Introduction] we added more examples to motivate why fairness is important or necessary in FL and to motivate the specific design choice for fairness via agent awareness, following the suggestions of reviewer MUHj.
- [Related work] we discussed more related work on fair FL via personalization,  following the suggestions of reviewer MUHj.
- [Section 3, Section 4] we swapped the notations of K and T and used T to denote the number of communication rounds,  following the suggestion of reviewer p3Qb.
- [Section 5] we added two more baselines. i.e., Ditto and CGSV, following the suggestions of reviewer MUHj.
- [Section 5] we added the existing fair FL metric, i.e., agnostic loss, following the suggestions of reviewer wM5N.
- [Appendix A.1, Table 3] we reported the performance and fairness of different methods under 100 clients on MNIST to study the scalability of FOCUS, as suggested by reviewer wM5N.
- [Appendix A.2, Figure 2] we reported the test loss and test accuracy of different methods over FL communication rounds on MNIST and CIFAR to study the convergence of FOCUS, as suggested by reviewer wM5N.
- [Appendix A.3, Table 4] we reported the computation time for different fair metrics on MNIST under 10 and 100 clients to study the scalability and efficiency of the proposed metric FAA, following the suggestions of reviewer wM5N.
- [Appendix A.3, Table 5, Table 6] we reported the number of communication rounds that each method takes to achieve targeted accuracy on MNIST and CIFAR to study the computation costs of FOCUS, as suggested by reviewer wM5N.
- [Appendix A.3, Table 7] we reported the training time and inference time of different methods, as suggested by reviewer wM5N.
- [Appendix A.4, Table 8] we constructed a new evaluation by combining the clustering and Fedavg together as another baseline and compared it to FOCUS, as suggested by reviewer wM5N.
- [Appendix A.5, Table 9] we studied the effect of the different number of clusters $M$ in FOCUS, as suggested by reviewer p3Qb.

All updates are highlighted in blue in our revision. Please also let us know if there are other questions, and we really look forward to the discussion with the reviewers to further improve our paper. Thank you!

---

### Decision · Program_Chairs · 2023-01-20

**Decision:**

Reject

**Justification For Why Not Higher Score:**

See the aforementioned main concerns (A), (B), and (C).

**Justification For Why Not Lower Score:**

N/A.

**Metareview: Summary, Strengths And Weaknesses:**

The main contribution of this work lies in proposing a fairness metric based on the difference between the maximal and minimal per-client excess risk for fair federated learning, which the reviewers have acknowledged to be novel.

After reviewing and responding to the authors' rebuttal, the reviewers have also raised the following main concerns:

(A) Theoretical analysis involves only 2 clusters.

(B) The number of clusters needs to be known in advance. The authors' further clarification to this issue was a rather simplistic experiment showing that if you pick more clusters, it wouldn't hurt. However, in a more realistic example, how can we know how many clusters would be enough? If we keep increasing the number of clusters, what does it compromise?

(C) The lack of consideration of different fairness metrics in their empirical evaluation and explanations for the corresponding experimental results, for which we strongly encourage the authors to incorporate in their revised paper. Furthermore, I would encourage the authors to adopt a more appropriate perspective that no one fairness metric "rules": Being fair in one metric does not imply it is fair in another.

(D) There were some confusing claims in the paper and rebuttal that were eventually clarified. For example, the authors are referring to heterogeneity of data quality/contribution in terms of Bayes optimal loss, instead of heterogeneity of data distributions (as described in some parts of the paper). The authors are advised to revise to the former and avoid making references to the data distribution to eliminate confusion.
The authors' mention that each agent should "gain the same" from participating in federated learning added to the confusion. Fortunately, the discussion on absolute vs. relative performance perspective really helps us to understand the equity vs. fairness perspective of this work. We were initially confused by why the authors say "fairness is usually defined as the protection of a specific attribute", but it was later clarified that this does not refer to their work.

We encourage the authors to revise their work based on the reviewers' suggestions.